SciPost Physics

Submission

# Josephson current through the SYK model

Luca Dell'Anna

Dipartimento di Fisica e Astronomia "G. Galilei" e Sezione INFN, Università di Padova, via F. Marzolo 8, I-35131, Padova, Italy

## Abstract

We calculate the equilibrium Josephson current through a disordered interacting quantum dot described by a Sachdev-Ye-Kitaev model fully contacted by two BCS superconductors, such that all modes of the dot contribute to the coupling, which encodes hopping and spin-flip processes. We show that, at zero temperature and at the conformal limit, i.e. in the strong interacting limit, the Josephson current is suppressed by $U$, the strength of the interaction, as $\ln(U)/U$ and becomes universal, namely it gets independent on the superconducting gap. At finite temperature, instead, it depends on the ratio between the gap and the temperature. A proximity effect exists but the self-energy corrections induced by the coupling with the superconducting leads seem subleading as compared to the interaction self-energy and the tunneling matrix for large number of particles. Finally we compare the results of the original four-fermion model with those obtained considering zero interaction, two-fermions and a generalized $q$-fermion model.

# 1  Introduction

The Sachdev-Ye-Kitaev (SYK) model, a non-Fermi liquid describing fermions with infinite-range interactions, has attracted a lot of scientific interest in recent years [1, 2]. Compared to ordinary Fermi liquids, it displays highly unusual properties; for example, its resistivity is linear in temperature [3–5]. Moreover, it has been shown that the SYK model is dual to an anti-de Sitter space in two-dimensions  [2, 6–8], opening an alternative way for investigating black holes. Regarding its experimental implementation, several proposals have already been formulated in solid state physics based on quantum dots coupled to topological superconducting wires [9], graphene flakes with irregular boundary [10, 11], by applying optical lattices [12], and in the field of cavity quantum electrodynamics [13].

A lot of theoretical studies have been devoted to study many peculiar properties of the SYK model, either at equilibrium and out-of equilibrium. Different investigations about many aspects of this model have been carried out, ranging from the dynamics triggered by a quantum quench [14], to realizing traversable wormholes [15], or about the Bekenstein-Hawking entropy [16, 17] and the existence of anomalous power laws in the temperature dependent conductance [18, 19]. Several studies have been also conducted investigating the mesoscopic physics by the SYK model, considering a lattice of SYK dots [20], analyzing the thermoelectric transport [21] and the charge transport by coupling with metallic leads [22, 23], characterizing the current and supercurrent driven by double contact setups [24], looking at the dynamics by coupling with Markovian reservoirs [25], and thermal baths, detecting some peculiar thermalization properties [26–29]. Several attempts have been done also to include superconductivity in the SYK model [30–33] with the need of upgrading the original complex model to a spin-full version with, in addition, a mechanism of particle attraction provided by phonons or by a negative Hubbard term.

Despite this intense scientific activity done on the transport properties of the SYK model, one of the most interesting and yet-little studied topic is about the currents driven by superconducting leads. Non-equilibrium currents triggered by a voltage applied either through normal and superconducting leads have been investigated [24], however a study of the equilibrium Josephson current is still laking. The Josephson effect provides a fundamental signature of phase-coherent transport through mesoscopic sample [34]. We calculate the direct Josephson current obtained by contacting a SYK dot by two conventional Bardeen-Cooper-Schrieffer (BCS) superconductors. The coupling between the dot and the superconductors involves uniformly all the degrees of freedom of the dot and encodes either hooping and spin-flip processes, in the same spirit of what done linking a topological p-wave superconductor with s-wave superconducting leads [35, 36]. We show that a proximity effect is induced in the dot, which causes the tunneling current originated by a phase-difference without applying any voltage, however the self-energy of the dot is weakly affected by the coupling with the superconducting leads in the so-called conformal limit, namely for large interaction and in the limit of large number of particles. We found that, in this limit, the Josephson current is suppressed by $U$, the strength of the interaction, as $\ln(U)/U$ and becomes universal, namely the current gets independent on the superconducting pairing. This means that the Josephson current, at zero temperature, and in the conformal limit, is the same for all BCS-like superconductors. Strikingly this result turns out to be formally the same as that obtained for a chaotic Josephson junction in the ergodic and long-dwell time regime reported in Ref. [37]. At finite temperature $T$, instead, the dependence on the superconducting gap $\Delta$ is restored. The current turns out to be dependent on the ratio between the gap and the temperature and goes as $\Delta^2/T^2$ for large temperatures. Moreover we also considered the SYK model with two fermionic operators, keeping the coupling with the leads the same. In this case the current depends on $\Delta$ and is highly non sinusoidal. Finally for a generalized $q$-fermion SYK model, the SYK$_q$ model, we find that, for $q > 4$ and in the weak coupling regime, the current looses completely its dependence on the gap.

## 2 Model

We study a system composed by a dot, modeled by a complex SYK Hamiltonian $H_d$, and two-superconducting leads described by $H_0$. The hybridization of the dot and the superconducting reservoirs takes place by the tunneling term $H_T$. The full Hamiltonian is, therefore,

$$H = H_0 + H_T + H_d \tag{1}$$

where the first term

$$H_0 = \sum_{p=L,R} \left( \sum_{k,\sigma} (\epsilon_k - \mu_p) c^\dagger_{p\sigma k} c_{p\sigma k} + \sum_k \Delta_p e^{i\phi_p} c^\dagger_{p\uparrow k} c^\dagger_{p\downarrow -k} \right) + h.c. \tag{2}$$

describes the two BCS-Hamiltonians, contacted to the left side ($p = L$) and to the right side ($p = R$) of the dot, $c_{L\sigma k}$, $c_{R\sigma k}$ the annihilation and $c^\dagger_{L\sigma k}$, $c^\dagger_{R\sigma k}$ creation fermionic operators, $\epsilon_k$ is the single particle spectrum, $\mu_L$ and $\mu_R$ the chemical potentials, $\Delta_L$ and $\Delta_R$ the gaps with phases $\phi_L$ and $\phi_R$, respectively.

The second term is the tunneling Hamiltonian

$$H_T = \frac{1}{\sqrt{N}} \sum_{p=L,R} \sum_{k,\sigma,n} t_{pn\sigma} c^\dagger_{p\sigma k} d_n + h.c. \tag{3}$$

where $t_{Ln\sigma}$ and $t_{Rn\sigma}$ are the spin-dependent hybridization parameters. This tunneling Hamiltonian encodes either the hopping which allows a fermion to jump into or out of the dot with same spin projection and the spin-flip processes at the interface for opposite spin projections, in the same spirit of Refs. [35], [36], where a topological superconductor made of spinless fermions is coupled to s-wave BCS superconducting electrodes. The fermionic operators $d_n$ and $d^\dagger_n$ are defined in the dot. The last term of Eq. (1) is the following complex SYK Hamiltonian of the dot

$$H_d = \frac{1}{(2N)^{3/2}} \sum_{i,j,n,l=1}^{N} U_{ijnl} \, d^\dagger_i d^\dagger_j d_n d_l - \mu \sum_i d^\dagger_i d_i \tag{4}$$

where $N$ fermions have a disordered all-to-all four-body interaction $U_{ijnl}$, Gaussian distributed.

### 2.1 Tunneling term

Let us first consider the leads and the tunneling term. We introduce the following Nambu-Jona-Lasinio spinors

$$\Psi_{pk} = \begin{pmatrix} c_{p\uparrow k} \\ c^\dagger_{p\downarrow -k} \end{pmatrix}, \qquad \bar{\Psi}_{pk} = \begin{pmatrix} c^\dagger_{p\uparrow k} & c_{p\downarrow -k} \end{pmatrix} \tag{5}$$

for the fermions of the leads, and

$$D_n = \begin{pmatrix} d_n \\ d^\dagger_n \end{pmatrix}, \qquad \bar{D}_n = \begin{pmatrix} d^\dagger_n & d_n \end{pmatrix} \tag{6}$$

for the fermions of the dot. The Hamiltonian of the leads, $H_0$, in this representation, becomes

$$H_0 = \sum_{pk} \bar{\Psi}_{pk} \left[ (\epsilon_k - \mu_p) \tau_3 + \Delta_p \cos(\phi_p) \tau_1 - \Delta_p \sin(\phi_p) \tau_2 \right] \Psi_{pk} \tag{7}$$

where $\tau_1, \tau_2, \tau_3$ are Pauli matrices, and the tunneling Hamiltonian $H_T$ reads

$$H_T = \frac{1}{\sqrt{N}} \sum_{p,k,n} \left( \bar{\Psi}_{pk} \hat{T}_{pn} D_n + \bar{D}_n \hat{T}^\dagger_{pn} \Psi_{pk} \right) \tag{8}$$

where

$$\hat{T}_{pn} = \begin{pmatrix} t_{pn\uparrow} & 0 \\ 0 & -t^*_{pn\downarrow} \end{pmatrix}, \tag{9}$$

Let us take $\mu_R = \mu_L = \mu$, i.e. at equilibrium. Defining

$$G_{kp}^{-1} = i\omega + \xi_k \tau_3 + \Delta_p \cos(\phi_p)\tau_1 - \Delta_p \sin(\phi_p)\tau_2 \tag{10}$$

where $\xi_k = \epsilon_k - \mu$, and integrating over $\Psi$,

$$e^{-S_c} = \int \mathcal{D}\bar{\Psi}\mathcal{D}\Psi \exp\left\{-\sum_{pk\omega}\left[\bar{\Psi}_{pk}G_{pk}^{-1}\Psi_{pk} + \frac{1}{\sqrt{N}}\sum_n(\bar{\Psi}_{pk}\hat{T}_{pn}D_n + \bar{D}_n\hat{T}_{pn}^\dagger\Psi_{pk})\right]\right\} \tag{11}$$

we get the contribution to the action of the dot due to the coupling with the leads

$$S_c = -\sum_{nm\omega}\bar{D}_n(\omega)\mathcal{T}_{nm}(\omega)D_m(\omega) \tag{12}$$

where, after defining

$$\tilde{\Gamma}^+_{pnm} = \frac{1}{2}\left(t^*_{pn\uparrow}t_{pm\uparrow} + t_{pn\downarrow}t^*_{pm\downarrow}\right) \tag{13}$$

$$\tilde{\Gamma}^-_{pnm} = \frac{1}{2}\left(t^*_{pn\uparrow}t_{pm\uparrow} - t_{pn\downarrow}t^*_{pm\downarrow}\right) \tag{14}$$

$$\tilde{\Gamma}^{s+}_{pnm} = \frac{1}{2}\left(t^*_{pn\uparrow}t^*_{pm\downarrow} + t_{pn\downarrow}t_{pm\uparrow}\right) \tag{15}$$

$$\tilde{\Gamma}^{s-}_{pnm} = \frac{1}{2}\left(t^*_{pn\uparrow}t^*_{pm\downarrow} - t_{pn\downarrow}t_{pm\uparrow}\right) \tag{16}$$

the kernel reads

$$\mathcal{T}_{nm}(\omega) = \frac{1}{N}\sum_{kp}\left\{\frac{i\omega\left(\tau_0\tilde{\Gamma}^+_{pnm} + \tau_3\tilde{\Gamma}^-_{pnm}\right) - \xi_k\left(\tau_3\tilde{\Gamma}^+_{pnm} + \tau_0\tilde{\Gamma}^-_{pnm}\right)}{\xi_k^2 + \Delta_p^2 + \omega^2}\right.$$
$$\left. + \frac{\Delta_p\cos(\phi_p)\left(\tau_1\tilde{\Gamma}^{s+}_{pnm} + i\tau_2\tilde{\Gamma}^{s-}_{pnm}\right) - \Delta_p\sin(\phi_p)\left(\tau_2\tilde{\Gamma}^{s+}_{pnm} - i\tau_1\tilde{\Gamma}^{s-}_{pnm}\right)}{\xi_k^2 + \Delta_p^2 + \omega^2}\right\} \tag{17}$$

We can integrate over $\xi_k$, and introducing $\nu_0$ the density of states at the Fermi energy, equal for both sides, we get

$$\mathcal{T}_{nm}(\omega) = \frac{1}{N}\sum_p \frac{\pi\nu_0}{\sqrt{\omega^2 + \Delta_p^2}}\left\{i\omega\left(\tau_0\tilde{\Gamma}^+_{pnm} + \tau_3\tilde{\Gamma}^-_{pnm}\right) + \Delta_p\cos(\phi_p)\left(\tau_1\tilde{\Gamma}^{s+}_{pnm} + i\tau_2\tilde{\Gamma}^{s-}_{pnm}\right)\right.$$
$$\left. -\Delta_p\sin(\phi_p)\left(\tau_2\tilde{\Gamma}^{s+}_{pnm} - i\tau_1\tilde{\Gamma}^{s-}_{pnm}\right)\right\} \tag{18}$$

Defining, in the symmetric case, $\phi_L = -\phi_R = \phi/2$, $\Delta_L = \Delta_R = \Delta$, $t_{Rn\sigma} = t_{Ln\sigma} = t_{n\sigma}$ and

$$\Gamma^\pm_{nm} = 2\pi\nu_0\tilde{\Gamma}^\pm_{Rnm} = 2\pi\nu_0\tilde{\Gamma}^\pm_{Lnm} \tag{19}$$

$$\Gamma^{s\pm}_{nm} = 2\pi\nu_0\tilde{\Gamma}^{s\pm}_{Rnm} = 2\pi\nu_0\tilde{\Gamma}^{s\pm}_{Lnm} \tag{20}$$

summing over $p = R, L$, namely summing the right and left terms, we get

$$\mathcal{T}_{nm}(\omega) = \frac{1}{N}\frac{i\omega\left(\tau_0\Gamma^+_{nm} + \tau_3\Gamma^-_{nm}\right)}{\sqrt{\omega^2 + \Delta^2}} + \frac{1}{N}\frac{\Delta\cos(\phi/2)\left(\tau_1\Gamma^{s+}_{nm} + i\tau_2\Gamma^{s-}_{nm}\right)}{\sqrt{\omega^2 + \Delta^2}} \tag{21}$$

Let us now make the reasonable assumption that $t_{n\sigma} = t_{m\sigma} = t_\sigma$ for any $n$ and $m$, then we define $\Gamma^+_{nm} \equiv \Gamma_0 J_{nm}$, $\Gamma^-_{nm} \equiv \Gamma_3 J_{nm}$, $\Gamma^{s+}_{nm} \equiv \Gamma_1 J_{nm}$ and $i\Gamma^{s-}_{nm} \equiv \Gamma_2 J_{nm}$, where $J$ is a $N \times N$ unit matrix, a matrix consisting of all 1s

$$
J = \begin{pmatrix} 1 & 1 & \dots & 1 \\ 1 & 1 & \dots & 1 \\ \vdots & \vdots & \ddots & \vdots \\ 1 & 1 & \dots & 1 \end{pmatrix}
$$

Specifically

$$
\Gamma_0 = \pi v_0 (|t_\uparrow|^2 + |t_\downarrow|^2), \quad \Gamma_3 = \pi v_0 (|t_\uparrow|^2 - |t_\downarrow|^2), \quad \Gamma_1 = 2\pi v_0 \, \mathrm{Re}[t_\uparrow t_\downarrow], \quad \Gamma_2 = 2\pi v_0 \, \mathrm{Im}[t_\uparrow t_\downarrow] . \tag{22}
$$

For real values of $t_{pn\sigma}$ we have $\Gamma_2 = 0$. We can write, therefore,

$$
\mathcal{T}(\omega) = \frac{1}{N} \left( \frac{i\omega\Gamma_0}{\sqrt{\omega^2 + \Delta^2}} \tau_0 + \frac{i\omega\Gamma_3}{\sqrt{\omega^2 + \Delta^2}} \tau_3 + \frac{\Gamma_1 \Delta \cos(\phi/2)}{\sqrt{\omega^2 + \Delta^2}} \tau_1 + \frac{\Gamma_2 \Delta \cos(\phi/2)}{\sqrt{\omega^2 + \Delta^2}} \tau_2 \right) \tag{23}
$$

such that

$$
\mathcal{T}_{nm}(\omega) = \mathcal{T}(\omega) J_{nm}. \tag{24}
$$

Strictly, in order to take into account that the anomalous terms (proportional to $\tau_1$ and $\tau_2$) cannot have diagonal elements in the mode space which actually give zeros contributions in the action Eq. (12), we can use the following form

$$
\mathcal{T}'_{nm}(\omega) = \mathcal{T}_{nm}(\omega) - \frac{1}{N} \left( \frac{\Gamma_1 \Delta \cos(\phi/2)}{\sqrt{\omega^2 + \Delta^2}} \tau_1 + \frac{\Gamma_2 \Delta \cos(\phi/2)}{\sqrt{\omega^2 + \Delta^2}} \tau_2 \right) \delta_{nm} \tag{25}
$$

Actually either Eq. (24) and Eq. (25) give the same action Eq. (12), while by using Eq. (25) instead of Eq. (24) one gets subleading irrelevant terms for the current in the large $N$ limit (as shown in Appendix A), therefore, at leading orders, these expressions are equivalent.

Finally, let us discuss the role of the random fluctuations in the tunneling matrix. Let us suppose that $t_{n\sigma}$ are independent random variables whose distribution has $\sigma_\sigma$ as standard deviation, then we can define $t_\sigma \equiv \frac{1}{N} \sum_n t_{n\sigma}$, the average value, providing that $t_\sigma \neq 0$, namely in the presence of a residual coherence which makes it non-vanishing, which is still a random variable with standard deviation of the mean $\bar{\sigma}_\sigma = \frac{1}{\sqrt{N}} \sigma_\sigma$. As a result the hybridization matrices fluctuate statistically as $\Gamma_{s,nm} \simeq \Gamma_s J_{nm} + \frac{\delta\Gamma_{nm}}{\sqrt{N}}$ with some random variables $\delta\Gamma_{nm}$ with zero average. Random fluctuations in the tunneling, therefore, can be neglected for large $N$, which is the limit we are interested in and where SYK models can be treated analytically. Actually the terms of order $1/N$ for uniform matrices are marginally relevant which leads to finite contributions also for $N \to \infty$, while higher order terms are subleading and vanish upon increasing $N$. Since we are going to consider $N \gg 1$, we can neglect the effects of these random fluctuations. At finite $N$ instead, those effects should be taken into account, on the same footing of other sources of finite $N$ corrections. The previous argument can be of some relevance in the presence of a residual coherence. However, as we will be seeing in Sec. 4.2, at least at the leading order in the tunneling parameters, the Josephson current will depend only on the absolute values of the tunneling amplitudes while the phase fluctuations are irrelevant, obtaining the same result as that for the uniform tunneling matrix, providing that the uniform amplitudes are $|t_\sigma|^2 = \frac{1}{N} \sum_n |t_{n\sigma}|^2$.

Quite in general one can take somehow random fluctuations into account after defining the random distributions $P_\sigma(t_\sigma)$ and getting the distributions for the transmission parameters as $\rho_s(\Gamma) = \int dt_\uparrow dt_\downarrow \delta(\Gamma - \Gamma_s(t_\uparrow, t_\downarrow)) P_\uparrow(t_\uparrow) P_\downarrow(t_\downarrow)$, where $\Gamma_s(t_\uparrow, t_\downarrow)$ are defined above Eq. (23). In this way one can easily incorporate the randomness from the couplings between the dot and the leads, by integrating the observables (i.e. the Josephson current) over the $\Gamma_s$ with weights $\rho_s$, as done in Ref. [38].

## 3  SYK$_4$ Dot

The Hamiltonian of the dot is given by Eq. (4), where $U_{ijnl}$ are complex, independent Gaussian random couplings with zero mean obeying $U_{ijnl} = -U_{jinl} = U_{ijln}$, $U_{ijnl} = U^*_{nlij}$ and mean value $\overline{|U_{ijnl}|^2} = U^2$. Introducing $r$ replicas, $a = 1, \dots, r$, labeling the field as $d_{na}$, we can average over disorder so that the action of the uncoupled dot can be written as follows

$$S'_d = \sum_{n,a} \int_0^\beta d\tau \, d^\dagger_{na}(\tau) \, (\partial_\tau - \mu) \, d_{na}(\tau) - \frac{U^2}{4N^3} \sum_{a,b} \int_0^\beta d\tau d\tau' \left| \sum_n d^\dagger_{na}(\tau) d_{nb}(\tau') \right|^4 + S_c \quad (26)$$

### 3.1  Effective action

We can decouple the interaction, in different channels, introducing a number of auxiliary fields, $e^{-S'_d} = \int D\{Q^0 P^0 Q^P P Q^\Delta \Delta\} e^{-S_d}$, getting

$$S_d = \sum_{na} \int_0^\beta d\tau \, d^\dagger_{na}(\tau) \, (\partial_\tau - \mu) \, d_{na}(\tau) + \sum_{ab} \int_0^\beta d\tau d\tau' \Big[ \frac{N}{4c_0 U^2} [Q^0_{ab}(\tau, \tau')]^2 + \frac{N^3}{4c_1 U^2} |Q^P_{ab}(\tau, \tau')|^2$$

$$+ \frac{N^3}{4c_2 U^2} [Q^\Delta_{ab}(\tau, \tau')]^2 + \frac{N}{2} Q^0_{ab}(\tau, \tau') |P^{ab}_0(\tau, \tau')|^2 - Q^0_{ab}(\tau, \tau') P^{ab}_0(\tau, \tau') \sum_n d^\dagger_{na}(\tau) d_{nb}(\tau')$$

$$+ \frac{1}{4} Q^P_{ab}(\tau, \tau') \sum_{nm} P^{ab}_{nm}(\tau, \tau') P^{ab}_{mn}(\tau, \tau') - \frac{1}{2} Q^P_{ab}(\tau, \tau') \sum_{nm} d^\dagger_{na}(\tau) P^{ab}_{nm}(\tau, \tau') d_{mb}(\tau')$$

$$- \frac{1}{2} Q^P_{ab}(\tau, \tau') \sum_{nm} d^\dagger_{ma}(\tau) P^{ab}_{mn}(\tau, \tau') d_{nb}(\tau') + \frac{1}{2} Q^\Delta_{ab}(\tau, \tau') \sum_{nm} |\Delta^{ab}_{nm}(\tau, \tau')|^2 \quad (27)$$

$$- \frac{1}{2} Q^\Delta_{ab}(\tau, \tau') \sum_{nm} d^\dagger_{na}(\tau) \Delta^{ab}_{nm}(\tau, \tau') d^\dagger_{mb}(\tau') - \frac{1}{2} Q^\Delta_{ab}(\tau, \tau') \sum_{nm} d_{ma}(\tau) \Delta^{ab*}_{nm}(\tau, \tau') d_{nb}(\tau') \Big] + S_c$$

where the weights $c_0, c_1, c_2$ are arbitrary positive real numbers such that $c_0 + c_1 + c_2 = 1$. The auxiliary fields are such that $Q^\Delta_{ab}(\tau, \tau')$ is real, $\Delta^{ab}_{nm}(\tau, \tau')$ is complex and $\Delta^{ab}_{nm}(\tau, \tau') = \Delta^{ab}_{mn}(\tau, \tau')$, while $Q^P_{ab}(\tau, \tau')$ is complex and $Q^P_{ab}(\tau, \tau') = Q^{P*}_{ba}(\tau', \tau)$ and $P^{ab}_{nm}(\tau, \tau') = P^{ab}_{mn}(\tau, \tau')$ can be taken real (it can be complex but only the real part matters), while $Q^0_{ab}(\tau, \tau') = Q^0_{ba}(\tau', \tau)$ is real and $P^{ab}_0(\tau, \tau') = P^{ba*}_0(\tau', \tau)$ complex. Using the representation in Eq. (6), with replica indices, namely $\bar{D}^a_n = \left( d^\dagger_{na} \; d_{na} \right)$ and $D^b_m = \begin{pmatrix} d_{mb} \\ d^\dagger_{mb} \end{pmatrix}$, we can write

$$S_d = \frac{1}{2} \sum_{nmab} \int_0^\beta d\tau d\tau' \Big\{ \bar{D}^a_n(\tau) \Big[ \delta_{\tau\tau'} \delta_{ab} \delta_{nm} (\tau_0 \partial_\tau - \tau_3 \mu) - \frac{1}{2} Q^0_{ab}(\tau, \tau') \delta_{nm} \Big( P^{ab}_0(\tau, \tau')(\tau_3 + \tau_0)$$

$$+ P^{ab*}_0(\tau, \tau')(\tau_3 - \tau_0) \Big) - \frac{1}{2} \Big( Q^P_{ab}(\tau, \tau') P^{ab}_{nm}(\tau, \tau')(\tau_3 + \tau_0) + Q^P_{ba}(\tau', \tau) P^{ba}_{nm}(\tau', \tau)(\tau_3 - \tau_0) \Big)$$

$$- \frac{1}{2} Q^\Delta_{ab}(\tau, \tau') \Big( \Delta^{ab}_{nm}(\tau, \tau')(\tau_1 + i\tau_2) + \Delta^{ab*}_{nm}(\tau, \tau')(\tau_1 - i\tau_2) \Big) \Big] D^b_m(\tau') \Big\}$$

$$+ \sum_{ab} \int_0^\beta d\tau d\tau' \Big[ \frac{N}{4U^2} \Big( \frac{1}{c_0} [Q^0_{ab}(\tau, \tau')]^2 + \frac{N^2}{c_1} |Q^P_{ab}(\tau, \tau')|^2 + \frac{N^2}{c_2} [Q^\Delta_{ab}(\tau, \tau')]^2 \Big) \quad (28)$$

$$+ \frac{N}{2} Q^0_{ab}(\tau, \tau') |P^{ab}_0(\tau, \tau')|^2 + \frac{1}{4} Q^P_{ab}(\tau, \tau') \sum_{nm} (P^{ab}_{nm}(\tau, \tau'))^2 + \frac{1}{2} Q^\Delta_{ab}(\tau, \tau') \sum_{nm} |\Delta^{ab}_{nm}(\tau, \tau')|^2 \Big] + S_c$$

We have to remind that the terms $P_{nm}$ and $\Delta_{nm}$, should be finite only for $n \neq m$. Let us now calculate the main contributions to the partition function, deriving the saddle point equations.

## 3.2 Saddle point equations

Imposing $\delta S_d = 0$ under varying the auxiliary fields we derive the following saddle point equations

$$P_0^{ab}(\tau, \tau') = -\frac{1}{2N} \sum_n \text{Tr}\left(\langle D_n^b(\tau')\bar{D}_n^a(\tau)\rangle(\tau_3 - \tau_0)\right) \tag{29}$$

$$P_0^{ba}(\tau', \tau) = -\frac{1}{2N} \sum_n \text{Tr}\left(\langle D_n^b(\tau')\bar{D}_n^a(\tau)\rangle(\tau_3 + \tau_0)\right) \tag{30}$$

$$P_{nm}^{ab}(\tau, \tau') = -\frac{1}{2}\text{Tr}\left(\langle D_m^b(\tau')\bar{D}_n^a(\tau)\rangle(\tau_3 + \tau_0)\right) \tag{31}$$

$$P_{mn}^{ba}(\tau', \tau) = -\frac{1}{2}\text{Tr}\left(\langle D_m^b(\tau')\bar{D}_n^a(\tau)\rangle(\tau_3 - \tau_0)\right) \tag{32}$$

$$\Delta_{nm}^{ab}(\tau, \tau') = -\frac{1}{2}\text{Tr}\left(\langle D_m^b(\tau')\bar{D}_n^a(\tau)\rangle(\tau_1 - i\tau_2)\right) \tag{33}$$

$$\Delta_{nm}^{ab*}(\tau, \tau') = -\frac{1}{2}\text{Tr}\left(\langle D_m^b(\tau')\bar{D}_n^a(\tau)\rangle(\tau_1 + i\tau_2)\right) \tag{34}$$

$$Q_{ab}^0(\tau, \tau') = -c_0 U^2 \Bigg\{ |P_0^{ba}(\tau', \tau)|^2 + \frac{1}{2N} \sum_n \text{Tr}\Big[\langle D_n^b(\tau')\bar{D}_n^a(\tau)\rangle\big((\tau_3 + \tau_0)P_0^{ab}(\tau, \tau')$$
$$+ (\tau_3 - \tau_0)P_0^{ab*}(\tau, \tau'))\Big]\Bigg\} = c_0 U^2 |P_0^{ba}(\tau', \tau)|^2 \tag{35}$$

$$Q_{ab}^P(\tau, \tau') = -\frac{c_1 U^2}{N^3} \Bigg\{ \sum_{nm} (P_{nm}^{ba}(\tau', \tau))^2 + \sum_{nm} \text{Tr}\left(\langle D_m^b(\tau')\bar{D}_n^a(\tau)\rangle(\tau_3 - \tau_0)\right) P_{nm}^{ba}(\tau', \tau) \Bigg\}$$
$$= \frac{c_1 U^2}{N^3} \sum_{nm} (P_{nm}^{ba}(\tau', \tau))^2 \tag{36}$$

$$Q_{ba}^P(\tau', \tau) = -\frac{c_1 U^2}{N^3} \Bigg\{ \sum_{nm} (P_{nm}^{ab}(\tau, \tau'))^2 + \sum_{nm} \text{Tr}\left(\langle D_m^b(\tau')\bar{D}_n^a(\tau)\rangle(\tau_3 + \tau_0)\right) P_{nm}^{ab}(\tau, \tau') \Bigg\}$$
$$= \frac{c_1 U^2}{N^3} \sum_{nm} (P_{nm}^{ab}(\tau, \tau'))^2 \tag{37}$$

$$Q_{ab}^\Delta(\tau, \tau') = -\frac{c_2 U^2}{N^3} \Bigg\{ \sum_{nm} |\Delta_{nm}^{ba}(\tau', \tau)|^2 + \frac{1}{2} \sum_{nm} \text{Tr}\Big[\langle D_m^b(\tau')\bar{D}_n^a(\tau)\rangle\big((\tau_1 + i\tau_2)\Delta_{nm}^{ab}(\tau, \tau')$$
$$+ (\tau_1 - i\tau_2)\Delta_{nm}^{ab*}(\tau, \tau'))\Big]\Bigg\} = \frac{c_2 U^2}{N^3} \sum_{nm} |\Delta_{nm}^{ba}(\tau', \tau)|^2 \tag{38}$$

We restrict our attention to replica diagonal solutions, $P_0^{ab}(\tau, \tau') = \delta_{ab} G_0(\tau, \tau')$, $P_{nm}^{ab}(\tau, \tau') = \delta_{ab} G_{nm}(\tau, \tau')$ and $\Delta_{nm}^{ab}(\tau, \tau') = \delta_{ab} F_{nm}(\tau, \tau') = \delta_{ab} F_{nm}^*(\tau' - \tau)$. We define

$$\mathbf{G}_{nm}(\tau, \tau') = \begin{pmatrix} -G_0(\tau', \tau)\delta_{nm} + G_{nm}(\tau, \tau') & F_{nm}^*(\tau, \tau') \\ F_{nm}(\tau, \tau') & G_0(\tau, \tau')\delta_{nm} - G_{mn}(\tau', \tau) \end{pmatrix} \tag{39}$$

and the self-energies

$$\Sigma(\tau, \tau') = -\begin{pmatrix} Q^0(\tau, \tau')G_0(\tau, \tau') & 0 \\ 0 & -Q^0(\tau', \tau)G_0(\tau', \tau) \end{pmatrix} \tag{40}$$

$$L_{nm}(\tau, \tau') = -\begin{pmatrix} Q^P(\tau, \tau')G_{nm}(\tau, \tau') & 0 \\ 0 & -Q^P(\tau', \tau)G_{mn}(\tau', \tau) \end{pmatrix} \quad (41)$$

$$A_{nm}(\tau, \tau') = -\begin{pmatrix} 0 & Q^\Delta(\tau, \tau')F_{nm}(\tau, \tau') \\ Q^\Delta(\tau, \tau')F_{nm}^*(\tau, \tau') & 0 \end{pmatrix} \quad (42)$$

We can define

$$\mathcal{G}_0(\tau, \tau') = \begin{pmatrix} -G_0(\tau', \tau) & 0 \\ 0 & G_0(\tau, \tau') \end{pmatrix}, \quad (43)$$

$$\mathcal{G}_{nm}(\tau, \tau') = \begin{pmatrix} G_{nm}(\tau, \tau') & 0 \\ 0 & -G_{mn}(\tau', \tau) \end{pmatrix}, \quad (44)$$

$$\mathcal{F}_{nm}(\tau, \tau') = \begin{pmatrix} 0 & F_{nm}^*(\tau, \tau') \\ F_{nm}(\tau, \tau') & 0 \end{pmatrix} \quad (45)$$

At the saddle point, from Eqs. (29)-(34), we have

$$\mathbf{G}_{nm}(\tau, \tau') = \mathcal{G}_0(\tau, \tau')\delta_{nm} + \mathcal{G}_{nm}(\tau, \tau') + \mathcal{F}_{nm}(\tau, \tau') = -\langle D_m(\tau')\bar{D}_n(\tau)\rangle \quad (46)$$

which depends on the time difference $\bar{\tau} = \tau' - \tau \in [-\beta, \beta]$, namely $\mathbf{G}_{nm}(\tau, \tau') = \mathbf{G}_{nm}(\bar{\tau})$. In Fourier space the full matrix $\hat{\mathbf{G}}(\bar{\tau})$ in spinorial and in the multimodal spaces, including the tunneling contribution $\hat{\mathcal{T}}(\omega) = \mathcal{T}(\omega)J$, reads

$$\hat{\mathbf{G}}(\omega) = \left[ \left( (i\omega\tau_0 + \mu\tau_3 - \Sigma(\omega))\hat{\mathbb{I}} + \hat{\mathcal{T}}(\omega) - \left( \hat{L}(\omega) + \hat{A}(\omega) \right) \right]^{-1} \quad (47)$$

where, from Eqs. (35)-(38), the self-energies $\Sigma(\omega)$, $\hat{L}(\omega)$ and $\hat{A}(\omega)$ are the Fourier transforms of

$$\Sigma(\bar{\tau}) = c_0 U^2 \mathcal{G}_0(\bar{\tau})^2 \mathcal{G}_0(-\bar{\tau}) = -c_0 U^2 \begin{pmatrix} G_0(\bar{\tau})^2 G_0(-\bar{\tau}) & 0 \\ 0 & -G_0(-\bar{\tau})^2 G_0(\bar{\tau}) \end{pmatrix} \quad (48)$$

and of $\hat{L}(\bar{\tau})$ and $\hat{A}(\bar{\tau})$, whose elements are

$$L_{nm}(\bar{\tau}) = -\frac{c_1 U^2}{N^3} \sum_{kl} \mathcal{G}_{kl}(-\bar{\tau})^2 \mathcal{G}_{nm}(\bar{\tau}) = -\frac{c_1 U^2}{N^3} \sum_{kl} \begin{pmatrix} G_{kl}(-\bar{\tau})^2 G_{nm}(\bar{\tau}) & 0 \\ 0 & -G_{kl}(\bar{\tau})^2 G_{mn}(-\bar{\tau}) \end{pmatrix} \quad (49)$$

$$A_{nm}(\bar{\tau}) = -\frac{c_2 U^2}{N^3} \sum_{kl} \mathcal{F}_{kl}(\bar{\tau})^2 \mathcal{F}_{nm}(\bar{\tau}) = -\frac{c_2 U^2}{N^3} \sum_{kl} \begin{pmatrix} 0 & |F_{kl}(\bar{\tau})|^2 F_{nm}(\bar{\tau}) \\ |F_{kl}(\bar{\tau})|^2 F_{nm}^*(\bar{\tau}) & 0 \end{pmatrix} \quad (50)$$

One has to solve self-consistently Eqs. (47)-(50), fixing then $c_0, c_1, c_2$, with constraint $c_0 + c_1 + c_2 = 1$, by minimizing the action at the saddle point. However what we found is that, if $G_{nm}$ and $F_{mn} \sim 1/N^\delta$ with $\delta > 0$, the self-energies $\hat{L}$ and $\hat{A}$ can be neglected in the large $N$ limit. As we will see in Section 4.4, this seems to be the case.

# 4 Josephson current

As shown in the previous section, the self energies induced by he coupling can be neglected in the large $N$ limit. In such approximation the Green's function of the dot can be written as

$$\mathcal{G}_{nm}^{-1} = \mathcal{G}_0^{-1}\delta_{nm} + \mathcal{T}J_{nm} \quad (51)$$

where $\mathcal{G}_0$ is the Green's function of the uncoupled dot, solution of the equations

$$\mathcal{G}_0^{-1}(\omega) = i\omega\tau_0 + \mu\tau_3 - \Sigma(\omega) \quad (52)$$

$$\Sigma(\tau) = U^2 \mathcal{G}_0(\tau)^2 \mathcal{G}_0(-\tau) \quad (53)$$

Actually if we include $\mathcal{T}$ or $\mathcal{T}'$ in Eq. (52) we have just subleading corrections of order $O(1/N)$ in the diagonal self-energy $\Sigma(\omega)$, which can be neglected. Let us write the self-energy in the following form

$$\Sigma(\omega) = \Sigma_0(\omega)\tau_0 + \Sigma_3(\omega)\tau_3 \tag{54}$$

so that we can write

$$\mathcal{G}_0^{-1}(\omega) = \tilde{G}_0^{-1}(\omega)\tau_0 + \tilde{G}_3^{-1}(\omega)\tau_3 \equiv \left(i\omega - \Sigma_0(\omega)\right)\tau_0 + \left(\mu - \Sigma_3(\omega)\right)\tau_3 \tag{55}$$

Actually, from Eq. (43), defining

$$G_0(\omega) = \frac{1}{2}\int d\tau\, e^{i\omega\tau}\left(G_0(\tau) - G_0(-\tau)\right), \qquad G_3(\omega) = \frac{1}{2}\int d\tau\, e^{i\omega\tau}\left(G_0(\tau) + G_0(-\tau)\right) \tag{56}$$

we have that

$$\tilde{G}_0^{-1}(\omega) = \frac{G_0(\omega)}{G_0(\omega)^2 - G_3(\omega)^2}, \qquad \tilde{G}_3^{-1}(\omega) = \frac{G_3(\omega)}{G_3(\omega)^2 - G_0(\omega)^2} \tag{57}$$

The Josephson current can be obtain from the phase derivative of the free energy

$$I = -\frac{1}{\beta}\partial_\phi\sum_\omega \ln\left(\det[\mathcal{G}^{-1}(\omega)]\right) \tag{58}$$

where $\beta = 1/T$ is the inverse of the temperature and the determinant of $\mathcal{G}^{-1}(\omega)$, from Eq. (73), is given by

$$\det[\mathcal{G}^{-1}] = \left(\det[\mathcal{G}_0^{-1}]\right)^N\left(1 + N\operatorname{Tr}[\mathcal{T}\mathcal{G}_0] + N^2\frac{\det[\mathcal{T}]}{\det[\mathcal{G}_0^{-1}]}\right) \tag{59}$$

from which, using $\det[\mathcal{G}_0^{-1}] = (\tilde{G}_0^{-1})^2 - (\tilde{G}_3^{-1})^2$, we get the following expression

$$\det[\mathcal{G}^{-1}] = \left(\det[\mathcal{G}_0^{-1}]\right)^{N-1} \tag{60}$$
$$\times\left[\left(\tilde{G}_0^{-1}(\omega) + \frac{i\omega\,\Gamma_0}{\sqrt{\omega^2 + \Delta^2}}\right)^2 - \frac{(\Gamma_1^2 + \Gamma_2^2)\,\Delta^2\cos^2(\phi/2)}{\omega^2 + \Delta^2} - \left(\tilde{G}_3^{-1}(\omega) - \frac{i\omega\,\Gamma_3}{\sqrt{\omega^2 + \Delta^2}}\right)^2\right].$$

We observe that, since

$$\Gamma^2 \equiv \Gamma_1^2 + \Gamma_2^2 = 4\pi^2 v_0^2\left|t_\uparrow t_\downarrow\right|^2 \tag{61}$$

the coupling is finite for finite values of both $t_\uparrow$ and $t_\downarrow$, namely there should be finite values of both spin projections, therefore also spin-flip processes in the presence of strongly polarized fermions in the dot, in order to have a non-vanishing Josephson current.

From Eq. (58) we finally obtain the Josephson current

$$I = \frac{\sin(\phi)}{\beta}\sum_\omega\frac{\Gamma^2\Delta^2}{\Gamma^2\Delta^2\cos^2(\phi/2) - \left(\tilde{G}_0^{-1}(\omega)\sqrt{\omega^2 + \Delta^2} + i\omega\Gamma_0\right)^2 + \left(\tilde{G}_3^{-1}(\omega)\sqrt{\omega^2 + \Delta^2} - i\omega\Gamma_3\right)^2} \tag{62}$$

By numerically solving of Eqs. (52), (53), using Eqs. (54), (55), one gets the Josephson current for the SYK dot from Eq. (62). As a remark we point out that, using Eq. (25) instead of Eq. (24), we get the same expression for the current with subleading terms of order $O(1/N)$ (see Appendix A). The same observation is valid if we want to improve the bare SYK solution including $1/N$ corrections. Actually If we include those corrections in the bare Green's function, the phase independent part of the free energy acquires a term of order $O(1)$ but does not contribute to the Josephson current since it is phase independent, while the phase dependent part, which is $O(1)$, and therefore, the Josephson current, acquires trivially a term $O(1/N)$, which is subleading and vanishes for large $N$. As a result, the Josephson current remains the same in the large $N$ limit.

## 4.1 Large interaction limit

In the so-called conformal limit, namely for very large $U$, i.e. for $|\omega| \ll U$, the analytical solution of Eqs. (52) and (53), obtained for $\Sigma_3(0) = \mu$, implying $G_3(0) = 0$ and $\tilde{G}_0 = G_0$, and for $T \to 0$, is given by [1, 5, 16]

$$G_0^{-1}(\omega) = iC \, \text{sgn}(\omega)|\omega|^{1/2} \tag{63}$$

with $C = (U^2/\pi)^{1/4}$, solution of the equations $G_0^{-1}(\omega) = -\Sigma_0(\omega)$ and $\Sigma_0(\tau) = -U^2 G_0(\tau)^2 G_0(-\tau)$. The Josephson current, Eq. (62), for $T \to 0$, in the continuum, becomes

$$I = \frac{\Gamma^2 \Delta^2}{\pi} \sin(\phi) \int_0^\infty \frac{d\omega}{\Gamma^2 \Delta^2 \cos^2(\phi/2) + \left(C\sqrt{\omega(\omega^2 + \Delta^2)} + \omega\Gamma_0\right)^2 - \omega^2 \Gamma_3^2} \tag{64}$$

The term $\omega^2 \Gamma_3^2$ does not lead to any singularity since $\Gamma_0^2 - \Gamma_3^2 = 4\pi^2 v_0^2 \left|t_\uparrow t_\downarrow\right|^2$ which is exactly equal to $\Gamma^2$, Eq. (61), and is positive defined. This equation, for $U \gg \Gamma$, can be well approximated by

$$I \simeq \frac{\Gamma^2}{\pi} \sin(\phi) \int_0^\Delta \frac{d\omega}{\Gamma^2 \cos^2(\phi/2) + C^2 \omega} \tag{65}$$

getting the following analytical result

$$I \simeq \frac{\Gamma^2}{\pi C^2} \sin(\phi) \ln\left(1 + \frac{C^2 \Delta}{\Gamma^2 \cos^2(\phi/2)}\right) \tag{66}$$

For $U\Delta \gg \Gamma^2$, the current $I$ drops the dependence on $\Delta$, except for logarithmic corrections,

$$I \simeq \frac{1}{\sqrt{\pi}} \frac{\Gamma^2}{U} \sin(\phi) \ln\left(\frac{U\Delta}{\sqrt{\pi} \, \Gamma^2 \cos^2(\phi/2)}\right) \tag{67}$$

namely, we get a universal behavior, $I \sim \ln(\tilde{U})/\tilde{U}$, with $\tilde{U} = U/\Gamma^2$, that is valid for all BCS-like superconductors. Quite interestingly a very similar result for the Josephson current reported in Eq. (67), had been obtained for a disordered Josephson junction in the so-called ergodic regime with long dwell time, namely when the ergotic time is small compare to $\Delta^{-1}$ and the Thouless energy scale $E_T$ is such that $E_T \ll \Delta$ [37]. In our case $\Gamma^2/U$ plays the role of $E_T$, or alternatively $U/\Gamma^2$ the role of the dwell time, so that a large interaction $U$ corresponds to a large diffusion time in the dot. We remind that, contrary to the case reported in Ref. [37], where the dot is a cavity weakly linked to the superconducting leads and the coupling with the leads involves few modes of the dot, in the present paper we considered a dot fully contacted to the leads, namely all the modes contribute to the coupling. The long dwell time in Ref. [37] is due to the diffusion of the particles in the cavity while, in our case, it is due to strong correlations which produces such a sort of self-trapping phenomenon. Moreover the anomalous self-energy plays a crucial role in Ref. [37], while in our case this term is negligible for large $N$ because of the uniform coupling.

**Perturbative analysis:** Before we proceed, let us reconsider the previous case by expanding the free energy for small tunneling parameters $\Gamma_\alpha$, with $\alpha = 0, 1, 2, 3$. From Eq. (59), expanding in terms of $\Gamma_\alpha$, we have

$$\ln\left(\det[\mathcal{G}^{-1}]\right) = N \ln\left(\det[\mathcal{G}_0^{-1}]\right) + \ln\left(1 + N \, \text{Tr}[\mathcal{T}\mathcal{G}_0] + N^2 \frac{\det[\mathcal{T}]}{\det[\mathcal{G}_0^{-1}]}\right) \tag{68}$$

$$= N \ln\left(\det[\mathcal{G}_0^{-1}]\right) + N \, \text{Tr}[\mathcal{T}\mathcal{G}_0] + N^2 \frac{\det[\mathcal{T}]}{\det[\mathcal{G}_0^{-1}]} - \frac{1}{2} N^2 \left(\text{Tr}[\mathcal{T}\mathcal{G}_0]\right)^2 + o(\Gamma_\alpha^2)$$

where $\mathcal{T}$ is given by Eq. (23) and the bare Green's function is simply $\mathcal{G}_0 = G_0 \tau_0$, as before. As a result we have

$$\ln\left(\det[\mathcal{G}^{-1}]\right) = -2N \ln(G_0) + \frac{2iG_0\omega\Gamma_0}{\sqrt{\omega^2 + \Delta^2}} + \frac{G_0^2}{\omega^2 + \Delta^2}\left[(\Gamma_0^2 + \Gamma_3^2)\omega^2 - \Gamma^2\Delta^2\cos^2(\phi/2)\right] + o(\Gamma_\alpha^2) \quad (69)$$

where $\Gamma^2$ is given by Eq. (61). The Josephson current, Eq. (58), at the leading order in the tunneling parameters, then, reads

$$I = -\frac{\Gamma^2\Delta^2}{\beta}\sin(\phi)\sum_\omega \frac{G_0(\omega)^2}{\omega^2 + \Delta^2} + o(\Gamma_\alpha^2) \quad (70)$$

which can be obtained also expanding Eq. (62). At zero temperature, in the continuum, and in the conformal limit, using Eq. (63), we have

$$I \simeq \frac{\Gamma^2\Delta^2}{\sqrt{\pi}U}\sin(\phi)\int_\lambda^\infty \frac{d\omega}{\omega(\omega^2 + \Delta^2)} \simeq \frac{\Gamma^2}{\sqrt{\pi}U}\sin(\phi)\ln\left(\frac{\Delta}{\lambda}\right) \quad (71)$$

where we introduced a positive infrared cut-off $\lambda$ to guarantee the convergence of the integral. Quite interestingly Eq. (71) has the same form of Eq. (67), with $\lambda \propto \Gamma^2/U$. Actually, from first principles, $\lambda$ has to be a function of the the prefactor $\Gamma^2/U$ and viceversa, in such a way that when $\lambda \to 0$ also $\Gamma^2/U \to 0$, getting a vanishing current. Conversely, if $\lambda$ and $\Gamma^2/U$ were independent, there would be a possibility to reduce $\lambda$ getting absurdly an arbitrary large current at fixed, even weak, tunneling parameter or strong interaction. This dependence agrees with the fact that the tunneling and the spectral properties are related.

What we learned is that the leading order in the tunneling parameters is enough to catch the logarithmic form of the Josephson current in our system, also because the strong interaction limit is equivalent to the weak tunneling regime, $U \gg \Gamma$, validating the perturbative expansion.

## 4.2 Random couplings

Let us now introduce random fluctuations in the tunneling amplitudes, relaxing the uniform form for the tunneling matrix $\mathcal{T}_{nm}$.

**Random phases:**  We will consider first the case where random phases occur

$$t_{n\sigma} = |t_\sigma|\, e^{i\theta_n^\sigma} \quad (72)$$

where $\sigma = \uparrow, \downarrow$ and $\theta_n^\sigma$ can be a random angle depending on the orbital index of the SYK dot. Let us now redo the expansion for the free energy in the presence of a more general matrix $\mathcal{T}_{nm}$. Let us write

$$\mathcal{G}^{-1} = \mathcal{G}_0^{-1}\mathbb{I} + \hat{\mathcal{T}} \quad (73)$$

with $\hat{\mathcal{T}}$ a matrix whose elements are $\mathcal{T}_{nm}$ reported in Eq. (21), where, now, from Eqs. (13)-(16), (19), (20), and (72),

$$\Gamma_{nm}^+ = \frac{1}{2}\left(\Gamma^\uparrow e^{-i(\theta_n^\uparrow - \theta_m^\uparrow)} + \Gamma^\downarrow e^{i(\theta_n^\downarrow - \theta_m^\downarrow)}\right) \quad (74)$$

$$\Gamma_{nm}^- = \frac{1}{2}\left(\Gamma^\uparrow e^{-i(\theta_n^\uparrow - \theta_m^\uparrow)} - \Gamma^\downarrow e^{i(\theta_n^\downarrow - \theta_m^\downarrow)}\right) \quad (75)$$

$$\Gamma_{nm}^{s+} = \frac{\Gamma}{2}\left(e^{-i(\theta_n^\uparrow + \theta_m^\downarrow)} + e^{i(\theta_n^\downarrow + \theta_m^\uparrow)}\right) \quad (76)$$

$$\Gamma_{nm}^{s-} = \frac{\Gamma}{2}\left(e^{-i(\theta_n^\uparrow + \theta_m^\downarrow)} - e^{i(\theta_n^\downarrow + \theta_m^\uparrow)}\right) \quad (77)$$

with

$$\Gamma^\uparrow = 2\pi\nu_0\left|t_\uparrow\right|^2, \quad \Gamma^\downarrow = 2\pi\nu_0\left|t_\downarrow\right|^2, \quad \Gamma = 2\pi\nu_0\left|t_\uparrow t_\downarrow\right|, \quad (78)$$

noticing that $\Gamma^2 = \Gamma^\uparrow \Gamma^\downarrow$. The free energy, apart from $-\beta^{-1}$, then reads

$$\ln\left(\det[\mathcal{G}^{-1}]\right) = N \ln\left(\det[\mathcal{G}_0^{-1}]\right) + \text{Tr} \ln\left(\mathbb{I} + \mathcal{G}_0 \hat{\mathcal{T}}\right) \tag{79}$$

Expanding Eq. (79) in terms of the tunneling matrix, we obtain

$$\ln\left(\det[\mathcal{G}^{-1}]\right) = -2N \ln(G_0) + \sum_n \text{Tr}\left(G_0 \mathcal{T}_{nn}\right) - \frac{1}{2} \sum_{nm} \text{Tr}\left(G_0 \mathcal{T}_{nm} G_0 \mathcal{T}_{mn}\right) + o(\mathcal{T}^2) \tag{80}$$

Let us consider the terms separately. The first order term is given by

$$\sum_n \text{Tr}\left(G_0 \mathcal{T}_{nn}\right) = \frac{iG_0\omega}{\sqrt{\omega^2 + \Delta^2}}(\Gamma^\uparrow + \Gamma^\downarrow) = \frac{2iG_0\omega\,\Gamma_0}{\sqrt{\omega^2 + \Delta^2}} \tag{81}$$

which is exactly the same as that appearing in Eq. (69), since $\Gamma_0 = (\Gamma^\uparrow + \Gamma^\downarrow)/2$, from Eq. (22). The second order term is the one relevant for the Josephson current

$$-\frac{1}{2} \sum_{nm} \text{Tr}\left(G_0 \mathcal{T}_{nm} G_0 \mathcal{T}_{mn}\right) = \frac{1}{N^2} \frac{G_0^2}{\omega^2 + \Delta^2} \sum_{nm} \left[\omega^2 \left(\Gamma_{nm}^+ \Gamma_{mn}^+ + \Gamma_{nm}^- \Gamma_{mn}^-\right)\right.$$
$$\left. -\Delta^2 \cos^2(\phi/2) \left(\Gamma_{nm}^{s+} \Gamma_{mn}^{s+} - \Gamma_{nm}^{s-} \Gamma_{mn}^{s-}\right)\right] \tag{82}$$

From Eqs. (74)-(77) we have

$$\Gamma_{nm}^\pm \Gamma_{mn}^\pm = \frac{1}{4}\left[\Gamma^{\uparrow 2} + \Gamma^{\downarrow 2} \pm 2\Gamma^\uparrow \Gamma^\downarrow \cos(\theta_m^\uparrow - \theta_n^\uparrow + \theta_m^\downarrow - \theta_n^\downarrow)\right] \tag{83}$$

$$\Gamma_{nm}^{s\pm} \Gamma_{mn}^{s\pm} = \frac{\Gamma^2}{2}\left[\cos(\theta_m^\uparrow + \theta_n^\uparrow + \theta_m^\downarrow + \theta_n^\downarrow) \pm 1\right] \tag{84}$$

therefore, intriguingly, in the combinations entering Eq. (82)

$$\Gamma_{nm}^+ \Gamma_{mn}^+ + \Gamma_{nm}^- \Gamma_{mn}^- = \frac{1}{2}\left(\Gamma^{\uparrow 2} + \Gamma^{\downarrow 2}\right) = \Gamma_0^2 + \Gamma_3^2 \tag{85}$$

$$\Gamma_{nm}^{s+} \Gamma_{mn}^{s+} - \Gamma_{nm}^{s-} \Gamma_{mn}^{s-} = \Gamma^2 \tag{86}$$

the phase dependence cancels out completely. As a result the second order term simplifies as follows

$$-\frac{1}{2} \sum_{nm} \text{Tr}\left(G_0 \mathcal{T}_{nm} G_0 \mathcal{T}_{mn}\right) = \frac{G_0^2}{\omega^2 + \Delta^2}\left[\left(\Gamma_0^2 + \Gamma_3^2\right)\omega^2 - \Gamma^2 \Delta^2 \cos^2(\phi/2)\right] \tag{87}$$

which is the same term appearing in Eq. (69), therefore, the Josephson current is also exactly the same as that reported in Eq. (71). In conclusion, replacing Eqs. (81) and (87) in Eq. (80) we get the same free energy obtained for a uniform phase, Eq. (69), and, then, the same current.

We proved, therefore, that, at least up to second order in the tunneling parameters, the random phases in the tunneling amplitudes do not play any role. At the same time we showed that the second order term is enough to derive the logarithmic form of the Josephson current, therefore the result in Eq. (71) is robust against random phase fluctuations.

**Fully random couplings:** Let us consider $t_{n\sigma}$ fully random variables so that the tunneling matrix has random entries $\mathcal{T}_{nm}$, Eq. (21), with tunneling parameters written in Eqs. (13)-(16), (19), (20), where we will drop the right-left index $p$, since we will consider symmetric contacts. Now let us consider once again the expansion of the free energy, which, up to second order in the tunneling parameters is given by Eq. (80). In this case the first order term reads

$$\sum_n \text{Tr}\left(G_0 \mathcal{T}_{nn}\right) = \frac{iG_0\omega}{\sqrt{\omega^2 + \Delta^2}} \frac{2\pi\nu_0}{N} \sum_n \left(|t_{n\uparrow}|^2 + |t_{n\downarrow}|^2\right) \equiv \frac{iG_0\omega}{\sqrt{\omega^2 + \Delta^2}}(\bar{\Gamma}^\uparrow + \bar{\Gamma}^\downarrow) \tag{88}$$

where we defined, analogously to Eq. (78), the following positive real quantities

$$\bar{\Gamma}^\uparrow = 2\pi\nu_0 \frac{1}{N}\sum_n \left|t_{n\uparrow}\right|^2, \qquad \bar{\Gamma}^\downarrow = 2\pi\nu_0 \frac{1}{N}\sum_n \left|t_{n\downarrow}\right|^2, \qquad \bar{\Gamma}^2 = \bar{\Gamma}^\uparrow\bar{\Gamma}^\downarrow. \tag{89}$$

The second order term is given by Eq. (82) where now, from Eqs. (13)-(16), (19), (20), we have

$$\Gamma_{nm}^+\Gamma_{mn}^+ + \Gamma_{nm}^-\Gamma_{mn}^- = 2\pi^2\nu_0^2 \left(\left|t_{n\uparrow}t_{m\uparrow}\right|^2 + \left|t_{n\downarrow}t_{m\downarrow}\right|^2\right) \tag{90}$$

$$\Gamma_{nm}^{s+}\Gamma_{mn}^{s+} - \Gamma_{nm}^{s-}\Gamma_{mn}^{s-} = 2\pi^2\nu_0^2 \left(\left|t_{n\downarrow}t_{m\uparrow}\right|^2 + \left|t_{n\uparrow}t_{m\downarrow}\right|^2\right) \tag{91}$$

As a result, under the double sum, for the second order term, using Eq. (89), we have

$$-\frac{1}{2}\sum_{nm}\text{Tr}\left(G_0\mathcal{T}_{nm}G_0\mathcal{T}_{mn}\right) = \frac{G_0^2}{\omega^2+\Delta^2}\left[\frac{1}{2}\left(\bar{\Gamma}^{\uparrow 2}+\bar{\Gamma}^{\downarrow 2}\right)\omega^2 - \bar{\Gamma}^2\Delta^2\cos^2(\phi/2)\right] \tag{92}$$

As a consequence the Josephson current is the same as in the uniform case, Eq. (71), where we replace $\Gamma \to \bar{\Gamma}$, namely $\Gamma^\sigma \to \bar{\Gamma}^\sigma$, or equivalently defining

$$|t_\sigma|^2 = \frac{1}{N}\sum_n |t_{n\sigma}|^2. \tag{93}$$

In conclusion, the Josephson current has exactly the same form for either uniform and completely random tunneling matrix, at least up to second order in the tunneling parameters, providing the equivalence given in Eq. (93).

This analysis shows that randomness in the dot-to-leads couplings does not spoil the current and justifies our choice of a simple uniform tunneling matrix which allowed us to perform an analytical calculation of the full Josephson current at any order.

## 4.3 Finite temperature

At finite temperature, the analytical solution Eq. (63) becomes [1, 5, 16]

$$G_0^{-1}(\omega) = iC\sqrt{2\pi T}\, e^{i\theta}\frac{\Gamma(3/4+\omega/2\pi T+i\epsilon)}{\Gamma(1/4+\omega/2\pi T+i\epsilon)} \tag{94}$$

where $\Gamma(x)$ is the Gamma function, $C = (U^2\cos(2\theta)/\pi)^{1/4}$, while $\theta$ and $\epsilon$ are linked by $e^{2\pi\epsilon} = \sin(\pi/4+\theta)/\sin(\pi/4-\theta)$, and $G_0(\tau=0^-) = 1/2 - \theta/\pi - \sin(2\theta)/4$. Let us fix the density of particles at half-filling, $\theta = 0$, $\epsilon = 0$. Defining

$$g_\omega = i\,\Gamma G_0(\omega) \tag{95}$$

from Eq. (62), in the case $UT \gg \Gamma^2$, the Josephson current becomes

$$I \simeq \frac{\Delta^2}{\beta}\sin(\phi)\sum_\omega \frac{g_\omega^2}{\omega^2 + g_\omega^2\Delta^2\cos^2(\phi/2) + \Delta^2} \tag{96}$$

The Green's function $G_0(\omega)$ is cut-offed by $1/\sqrt{T}$ at low frequency. We approximate, therefore, $g_\omega \approx g_0$ in Eq. (96) and, after summing over the Matsubara frequencies, we get

$$I \simeq \frac{\Delta}{2\alpha}\sin(\phi)\, g_0^2\, \frac{\tanh\left(\frac{\beta}{2}\Delta\sqrt{1+g_0^2\cos^2(\phi/2)}\right)}{\sqrt{1+g_0^2\cos^2(\phi/2)}} \tag{97}$$

which is a function of the temperature $T = 1/\beta$, and of the interaction $U$ since $g_0 = r\Gamma/\sqrt{UT}$, with $r$ a numerical coefficient, $r = \Gamma(1/4)/(\sqrt{2}\pi^{1/4}\Gamma(3/4))$. We find numerically that Eq. (96) is better approximated by the same expression where $g_\omega$ is replaced by $g_0$ if we include an overall factor $\alpha \approx 5.6$. Since $g_0^2 \ll 1$, calling $c = r^2/(2\alpha)$ the numerical coefficient, we have

$$I \simeq c\frac{\Gamma^2\Delta}{UT}\sin(\phi)\tanh\left(\frac{\Delta}{2T}\right) \tag{98}$$

therefore, for large temperature, $T \gg \Delta$, it reads

$$I \simeq \frac{c}{2}\frac{\Gamma^2\Delta^2}{U\,T^2}\sin(\phi) \tag{99}$$

namely, approaching the superconductive critical temperature $T_c$, it vanishes as $\frac{\Delta^2}{T^2} \propto \frac{T_c(T_c-T)}{T^2}$. On the contrary, in the intermediate regime with small enough temperatures, specifically for $\Delta \gg T \gg \Gamma^2/U$, we can approximate the hyperbolic tangent by one, getting a $1/T$ decay

$$I \simeq \frac{\Delta}{2\alpha}\sin(\phi)\frac{g_0^2}{\sqrt{1+g_0^2\cos^2(\phi/2)}} \simeq c\frac{\Gamma^2\Delta}{U\,T}\sin(\phi) \tag{100}$$

For $UT \ll \Gamma^2$ ($g_0 \gg 1$), instead, we have to distinguish two regions in frequency space, with $|\omega| < \Lambda_T$ and $|\omega| > \Lambda_T$, where $\Lambda_T \sim T$ is an energy cut-off below which $g_\omega \sim g_0$ while above $g_\omega \sim C^{-1}\mathrm{sgn}(\omega)|\omega|^{-1/2}$, as for the zero temperature limit. We have, therefore, the following expression

$$I \simeq \frac{1}{\beta}\sin(\phi)\left\{\sum_{|\omega|<\Lambda_T}\frac{\Delta^2}{\omega^2+\Delta^2\cos^2(\phi/2)+g_0^{-2}\Delta^2} + \sum_{\Delta>|\omega|>\Lambda_T}\frac{\Gamma^2}{\Gamma^2\cos^2(\phi/2)+g_\omega^{-2}}\right\} \tag{101}$$

Since $T \ll 1$ we can use the integrals, $\frac{1}{\beta}\sum_\omega \rightarrow \int\frac{d\omega}{2\pi}$, getting

$$I \simeq \frac{\Delta}{\pi}\sin(\phi)\,g_0\frac{\arctan\left(\frac{g_0\Lambda_T}{\sqrt{1+g_0^2\cos^2(\phi/2)}}\right)}{\sqrt{1+g_0^2\cos^2(\phi/2)}} + \frac{\Gamma^2}{\pi C^2}\sin(\phi)\ln\left(\frac{\Gamma^2\cos^2(\phi/2)+C^2\Delta}{\Gamma^2\cos^2(\phi/2)+C^2\Lambda_T}\right). \tag{102}$$

### 4.4 Proximity effect

Let us discuss, now, how the dot is affected by the presence of the superconducting leads and check whether we can neglect the self-energy corrections in the large $N$ limit. We will focus in particular on the hybridization of the dot due to the superconducting pairing, considering the following tunneling matrix, neglecting, for simplicity, the term proportional to $\tau_0$,

$$\hat{\mathcal{T}}(\omega) \simeq \frac{1}{N}\mathcal{T}_1(\omega)\,\tau_1\,J \equiv \frac{\Gamma\Delta\cos(\phi/2)}{N\sqrt{\omega^2+\Delta^2}}\,\tau_1\,J \tag{103}$$

We make the following ansatz for the anomalous contribution to the self-energy: $A\,\tau_1 J$. The Green's function, then, reads

$$\mathcal{G}_{nm}^{-1} \simeq G_0^{-1}\tau_0\,\delta_{nm} + \left(\frac{1}{N}\mathcal{T}_1(\omega) - A\right)\tau_1\,J_{nm} \tag{104}$$

From Eq. (33), we have the following effective equal-time pairing between two generic modes $n \neq m$

$$F \equiv F_{nm}(\tau,\tau) = \frac{1}{\beta}\sum_\omega \mathrm{Tr}\big(\mathcal{G}(\omega)\tau_1\big)_{nm} \tag{105}$$

and, therefore, from Eq. (50), we will have, consistently,

$$A = -\frac{U^2 F^3}{N} \tag{106}$$

At low temperature the sum in Eq. (105) becomes an integral, which reads

$$F = \int_{-\Lambda}^{\Lambda} \frac{d\omega}{2\pi} \frac{\mathcal{T}_1(\omega)/N - A}{N^2(\mathcal{T}_1(\omega)/N - A)^2 - G_0^{-2}} \tag{107}$$

and, using Eqs. (63), (103) and (106),

$$F = \frac{1}{N} \int_{-\Lambda}^{\Lambda} \frac{d\omega}{2\pi} \frac{\Gamma\Delta\cos(\phi/2)\sqrt{\omega^2 + \Delta^2} + U^2 F^3(\omega^2 + \Delta^2)}{\left(\Gamma\Delta\cos(\phi/2) + U^2 F^3\sqrt{\omega^2 + \Delta^2}\right)^2 + C^2|\omega|(\omega^2 + \Delta^2)} \tag{108}$$

where we introduced a cut-off since $\omega \ll U$ for the expression of $G_0$ to be valid, therefore we can take $\Lambda \sim U$.

For large $U$ and for large but still finite $N$ such that $U^2 F^3 \gg \Gamma$, we can approximate Eq. (132) getting

$$F \simeq \frac{1}{N} \int_0^{\Lambda} \frac{d\omega}{\pi} \frac{U^2 F^3}{(U^4 F^6 + C^2\omega)} = \frac{U^2 F^3}{\pi N C^2} \ln\left(1 + \frac{C^2\Lambda}{U^4 F^6}\right) \tag{109}$$

which has to be solved in terms of $F$. For $U^4 F^6 \gg C^2\Lambda \sim U^2$, we get, for $F$ and $A$, the following results

$$F \approx \left(\frac{\Lambda}{N\pi U^2}\right)^{1/4}, \qquad A \approx -\frac{U^{2/3}}{N}\left(\frac{\Lambda}{N\pi}\right)^{3/4} \tag{110}$$

We found that the pairing is super-extensive, meaning that a single particle in the dot is paired with all the other particles in such a way that $NF$ is not $O(1)$ but $O(N^{3/4})$.

We expect that $U^2 F^3$ becomes irrelevant upon further increasing $N$, therefore Eq. (132) becomes

$$F = \frac{1}{N} \int_{-\infty}^{\infty} \frac{d\omega}{2\pi} \frac{\Gamma\Delta\cos(\phi/2)\sqrt{\omega^2 + \Delta^2}}{\Gamma^2\Delta^2\cos^2(\phi/2) + C^2|\omega|(\omega^2 + \Delta^2)} \tag{111}$$

which can be approximated by

$$F \simeq \frac{\Gamma\Delta^2}{N\pi}\cos(\phi/2)\int_0^{\Lambda} \frac{d\omega}{\Gamma^2\Delta^2\cos^2(\phi/2) + C^2\Delta^2\omega} \simeq \frac{1}{N}\frac{\Gamma}{\pi C^2}\cos(\phi/2)\ln\left(1 + \frac{C^2\Lambda}{\Gamma^2\cos^2(\phi/2)}\right) \tag{112}$$

where now $\Lambda \sim \max(\Delta, \Gamma\cos(\phi/2))$. Therefore we have

$$F \sim \frac{1}{N}\frac{\Gamma}{U}\cos(\phi/2)\ln\left(\frac{U}{\Gamma\cos^2(\phi/2)}\right), \qquad A = -\frac{U^2 F^3}{N} \sim -\frac{(\Gamma\ln(U))^3}{U N^4} \tag{113}$$

This result implies that, even if the pairing is a uniform matrix whose elements are $\propto \frac{1}{N}$, the corresponding self-energy decays much faster upon increasing $N$, validating the approach used for calculating the Josephson current.

## 5 Other cases

Let us consider a couple of different situations in order to compare the results with those obtained for the SYK model. The first case is just the non-interacting dot with $N$ modes. The second one is the bilinear version of the SYK model, also called $SYK_2$ (the original model is also called $SYK_4$), that is a non-interacting all-to all random hopping model.

## 5.1  Zero interaction

For $U = 0$ we have $\Sigma = 0$, therefore $\tilde{G}_0^{-1} = i\omega$ and $\tilde{G}_3^{-1} = \mu$, therefore the Josephson current, Eq. (62), becomes

$$I = \frac{\Gamma^2 \Delta^2}{\beta} \sin(\phi) \sum_\omega \frac{1}{\Gamma^2 \Delta^2 \cos^2(\phi/2) + \left(\omega\sqrt{\omega^2 + \Delta^2} + \omega\Gamma_0\right)^2 + \left(\mu\sqrt{\omega^2 + \Delta^2} - i\omega\Gamma_3\right)^2} \tag{114}$$

For $\Gamma_0 \gg \Delta$, using $\Gamma_0^2 - \Gamma_3^2 = \Gamma^2$, and approximating Eq. (114) as follows

$$I \simeq \frac{\Gamma^2 \Delta^2}{\beta} \sin(\phi) \sum_\omega \frac{1}{\Gamma^2 \Delta^2 \cos^2(\phi/2) + \mu^2 \Delta^2 + \omega^2(\Gamma^2 + \mu^2) - 2i\omega\mu\Gamma_3\Delta} \tag{115}$$

we can sum over the Matsubara frequencies by complex analysis. Introducing the transmission coefficient, $t_o$ ranging from 0 to 1,

$$t_o = \frac{\Gamma^2}{\Gamma^2 + \mu^2} \tag{116}$$

we get the following analytical form

$$I \simeq \frac{\Delta}{2} \sin(\phi) \frac{t_o \sinh\left(\beta\Delta\sqrt{1 - t_o \sin^2(\phi/2) + t_o(1 - t_o)\Gamma_3^2/\Gamma^2}\right)}{\cosh\left(\beta\Delta\sqrt{1 - t_o \sin^2(\phi/2) + t_o(1 - t_o)\Gamma_3^2/\Gamma^2}\right) + \cosh\left(\beta\Delta\sqrt{t_o(1 - t_o)}\Gamma_3/\Gamma\right)}$$
$$\times \frac{1}{\sqrt{1 - t_o \sin^2(\phi/2) + t_o(1 - t_o)\Gamma_3^2/\Gamma^2}} \tag{117}$$

For $\Gamma_3 = 0$, namely for $|t_\uparrow| = |t_\downarrow|$, Eq. (117) reduces to the same result of a spinfull single level dot

$$I \simeq \frac{\Delta}{2} \sin(\phi) \frac{t_o \tanh\left(\frac{\beta}{2}\Delta\sqrt{1 - t_o \sin^2(\phi/2)}\right)}{\sqrt{1 - t_o \sin^2(\phi/2)}} \tag{118}$$

For $T \to 0$, Eq. (117) becomes simply

$$I \simeq \frac{\Delta}{2} \sin(\phi) \frac{t_o}{\sqrt{1 - t_o \sin^2(\phi/2) + t_o(1 - t_o)\Gamma_3^2/\Gamma^2}} \tag{119}$$

For large temperature, $T \gg \Delta$, the current in Eq. (117) becomes

$$I \simeq \frac{\Delta^2 t_o}{4T} \sin(\phi) \tag{120}$$

namely, it decays as $\Delta^2/T$. This result has to be contrasted with Eq. (99) obtained for large interaction.

## 5.2  SYK$_2$ dot

Let us consider the SYK version with two fermions, called SYK$_2$, which reads

$$H_d = \frac{1}{\sqrt{N}} \sum_{i,j=1}^N U_{ij} d_i^\dagger d_j \tag{121}$$

with $U_{ij}$ random variables whose mean value is $\overline{|U_{ij}|^2} = U^2$. In the strong coupling and for large $N$, at for zero temperature, the single particle Green's function has an analytic form

$$G_0^{-1}(\omega) = iU \, \mathrm{sgn}(\omega) \tag{122}$$

As done for the SYK$_4$ model, we can integrate over disorder getting

$$S_d' = \sum_{n,a} \int_0^\beta d\tau \, d_{na}^\dagger(\tau) \, (\partial_\tau - \mu) \, d_{na}(\tau) + \frac{U^2}{2N} \sum_{a,b} \int_0^\beta d\tau d\tau' \left| \sum_n d_{na}^\dagger(\tau) d_{nb}(\tau') \right|^2 + S_c \tag{123}$$

which can be decoupled as

$$S_d = \sum_{na} \int_0^\beta d\tau \, d_{na}^\dagger(\tau) \, (\partial_\tau - \mu) \, d_{na}(\tau) + \sum_{ab} \int_0^\beta d\tau d\tau' \left[ \frac{N}{2U^2} \left| \Sigma^{ab}(\tau,\tau') \right|^2 + \frac{N}{2U^2} \sum_{nm} \left| F_{nm}^{ab}(\tau,\tau') \right|^2 \right.$$

$$+ i\Sigma^{ba}(\tau',\tau) \sum_n d_{na}^\dagger(\tau) d_{nb}(\tau') - \frac{1}{2} \sum_{nm} \left( d_{na}^\dagger(\tau) F_{nm}^{ab*}(\tau,\tau') d_{mb}^\dagger(\tau') + d_{nb}(\tau') F_{nm}^{ab}(\tau,\tau') d_{ma}(\tau) \right) \left. \right] + S_c$$

with $\Sigma^{ab*}(\tau,\tau') = \Sigma^{ba}(\tau',\tau)$ and $F_{nm}^{ab}(\tau,\tau') = F_{mn}^{ab}(\tau,\tau')$. In the diagonal replica index, and for zero replica limit, the saddle point equations are

$$\Sigma(\tau,\tau') = -\frac{iU^2}{2N} \sum_n \langle d_n^\dagger(\tau) d_n(\tau') \rangle \tag{124}$$

$$F_{nm}(\tau,\tau') = \frac{U^2}{2N} \langle d_n^\dagger(\tau) d_m^\dagger(\tau') \rangle \tag{125}$$

One has to solve these equations self-consistently. Let us consider, for simplicity, $\mu = 0$, and $t_\sigma$ real, so that $\Gamma_2 = 0$, and $t_\uparrow = t_\downarrow$ which implies $\Gamma_3 = 0$. The tunneling matrix, therefore has only components proportional to $\tau_0$ and $\tau_1$. We then select only the self-energies in the same channels. The Green's function $\mathcal{G}$, then reads

$$\mathcal{G}_{nm}^{-1} = \mathcal{G}_0^{-1} \delta_{nm} + (\mathcal{T} + \hat{F}) J_{nm} = (i\omega - \Sigma)\tau_0 \, \delta_{nm} + \frac{1}{N} \left[ \mathcal{T}_0 \, \tau_0 + (\mathcal{T}_1 + NF) \, \tau_1 \right] J_{nm} \tag{126}$$

where we absorb $i$ in $\Sigma$, namely $i\Sigma \to \Sigma$, and consider uniform $\hat{F}$ like $\mathcal{T}$. We also define

$$\mathcal{T}_0 = \frac{i\omega \Gamma_0}{\sqrt{\omega^2 + \Delta^2}}, \qquad \mathcal{T}_1 = \frac{\Gamma_1 \Delta \cos(\phi/2)}{\sqrt{\omega^2 + \Delta^2}} \tag{127}$$

with $\Gamma_0 = \Gamma_1 \equiv \Gamma$ for our choice of the parameters $t_\sigma$. We can write the self-consistent equations, in Fourier space, in the following form

$$\Sigma = \frac{U^2}{2N} \sum_n \mathrm{Tr} \left( \mathcal{G}_{nn} \tau_0 \right) \tag{128}$$

$$F = \frac{U^2}{2N} \mathrm{Tr} \left( \mathcal{G}_{nm} \tau_1 \right) \tag{129}$$

We recognize in Eqs. (128), (129) the rainbow diagram equations reported in Refs. [39] and [37]. The self-energies can be obtained by employing Eq. (167) where we replace $\mathcal{G}_0'$ by $\mathcal{G}_0$ (we neglect corrections of order $1/N$ in the diagonal part, encoding Pauli exclusion principle, which would lead to subleading terms, as shown in Appendix A) and $\mathcal{T}$ by $(\mathcal{T} + \hat{F})$, namely

$$\mathcal{G}_{nm} = \mathcal{G}_0 \, \delta_{nm} - \left[ \left( N(\mathcal{T} + \hat{F}) + \mathcal{G}_0^{-1} \right)^{-1} (\mathcal{T} + \hat{F}) \mathcal{G}_0 \right] J_{nm} \tag{130}$$

The self-consistent equations Eqs. (128), (129), using Eq. (130), read explicitly

$$\Sigma = \frac{U^2}{(i\omega - \Sigma)}\left[1 - \frac{1}{N}\left(\frac{(i\omega - \Sigma + \mathcal{T}_0)\mathcal{T}_0 - (\mathcal{T}_1 + NF)^2}{(i\omega - \Sigma + \mathcal{T}_0)^2 - (\mathcal{T}_1 + NF)^2}\right)\right] \tag{131}$$

$$F = -\frac{U^2}{N}\left[\frac{(\mathcal{T}_1 + NF)}{(i\omega - \Sigma + \mathcal{T}_0)^2 - (\mathcal{T}_1 + NF)^2}\right] \tag{132}$$

In the large $N$ limit Eq. (131) reduces to the uncoupled self-energy $\Sigma = \frac{U^2}{(i\omega - \Sigma)}$ whose solution is

$$\Sigma = \frac{i}{2}\left(\omega - \text{sgn}(\omega)\sqrt{\omega^2 + 4U^2}\right) \tag{133}$$

which implies the following uncoupled dot Green's function

$$\mathcal{G}_0 = G_0\tau_0 = (i\omega - \Sigma)^{-1}\tau_0 = \frac{i}{2U^2}\left(\omega - \text{sgn}(\omega)\sqrt{\omega^2 + 4U^2}\right)\tau_0 \tag{134}$$

and for large $U$ one recovers Eq. (122), where $\text{sign}(\omega)$ comes from requiring vanishing $\Sigma$ in the zero interaction limit and then odd function in imaginary time. One can check that actually Eq. (134) is obtained by summing over the rainbow diagrams using the bare Green's function $(i\omega)^{-1}$

$$G_0 = \frac{1}{i\omega}\sum_{n=0}^{\infty} C_n \left(\frac{U}{i\omega}\right)^{2n} \tag{135}$$

where $C_n = \frac{(2n)!}{(n+1)\,n!}$ are the Catalan numbers. By inspection of Eq. (132) we have $F \sim 1/N$ therefore it can not be neglected in the large $N$ limit, since $NF$ appears in the Green's function. Defining for simplicity

$$\mathcal{A} = \mathcal{T}_1^2 + 3\left((i\omega - \Sigma + \mathcal{T}_0)^2 + U^2\right) \tag{136}$$

$$\mathcal{B} = 2\mathcal{T}_1^3 + 9\left(U^2 - 2(i\omega - \Sigma + \mathcal{T}_0)^2\right)\mathcal{T}_1 \tag{137}$$

we can solve algebraically Eq. (132) getting

$$NF = -\frac{2}{3}\mathcal{T}_1 + \frac{S}{3}\mathcal{A}\left(\frac{2}{\mathcal{B} + \sqrt{\mathcal{B}^2 - 4\mathcal{A}^3}}\right)^{1/3} + \frac{S^*}{3}\left(\frac{\mathcal{B} + \sqrt{\mathcal{B}^2 - 4\mathcal{A}^3}}{2}\right)^{1/3} \tag{138}$$

where $S = 1$, for $\mathcal{T}_1 > 0$ $(0 \le \phi \le \pi)$, and $S = -\frac{1+i\sqrt{3}}{2}$, for $\mathcal{T}_1 < 0$ $(\pi \le \phi \le 2\pi)$. Inserting Eq. (133) one can calculate the Josephson current

$$I = -\frac{1}{\beta}\sum_{\omega}\partial_\phi \ln\left[(\mathcal{T}_1 + NF)^2 - (i\omega - \Sigma + \mathcal{T}_0)^2\right] \tag{139}$$

which, since $\Sigma$ does not depend on $\phi$, can be also written as

$$I = -\frac{2}{U^2\beta}N\sum_{\omega} F\,\partial_\phi\,(\mathcal{T}_1 + NF) \tag{140}$$

Let us now consider the limit of very large $U$, namely $U \gg \Gamma$ and $\Delta$. In this limit $\Sigma \to -iU\text{sgn}(\omega)$, and $\mathcal{A} \to \mathcal{T}_1^2$, $\mathcal{B} \to 3^3U^2\mathcal{T}_1$, therefore

$$NF \simeq \text{sgn}(\cos(\phi/2))\frac{\left(U^2\Gamma\Delta\,|\cos(\phi/2)|\right)^{1/3}}{(\omega^2 + \Delta^2)^{1/6}} \tag{141}$$

Notice that this expression is accurate for $\phi$ far from $\pi$, since the term $\left((i\omega - \Sigma + \mathcal{T}_0)^2 + U^2\right)$ induces a gap close to $\phi = \pi$ for any finite value of $U$. The Josephson current is, therefore

$$I \simeq -\frac{1}{U^2\beta}N^2\sum_\omega \partial_\phi F^2 \tag{142}$$

since $NF \gg \mathcal{T}_1$. At zero temperature, introducing an ultraviolet cut-off $\Lambda$, we have

$$I \simeq \frac{\Delta}{6\pi}\left(\frac{\Gamma}{U}\right)^{2/3}\frac{\sin(\phi)}{(\cos^2(\phi/2))^{2/3}}\int_0^\Lambda \frac{d\omega}{(\omega^2+1)^{1/3}} \approx \frac{\Delta}{2\pi}\left(\frac{\Gamma}{U}\right)^{2/3}\frac{\sin(\phi)}{(\cos^2(\phi/2))^{2/3}}\Lambda^{1/3} \tag{143}$$

Since Eq. (122) is valid for $U \gg |\omega|$ the cut-off $\Lambda$ might be taken $\sim U$.

Here a comment is in order. If we used a different coupling with the leads, such that only few and specific modes of the dot were involved, the tunneling matrix would be sparse diagonal, with many null elements. In our case, since the dot is made by spinless fermions, stricktly speaking the anomalous terms should vanish. Nevertheless let us consider this case discussed in Ref. [37] where two superconducting leads were coupled to a chaotic metallic cavity. The calculation is pretty similar to what reported here. The only difference is that both $\mathcal{T}$ and $F$ are now diagonal. The saddle point equations are again equivalent to those obtained by resummation of the non-crossing diagrams, as shown also in our case. We have, then, to solve the self-consistent equations, where Eq. (126) is replaced by the following Eq. (146),

$$\Sigma = \frac{U^2}{2N}\sum_n \text{Tr}\left(\mathcal{G}_{nn}\tau_0\right) \tag{144}$$

$$F = \frac{U^2}{2N}\sum_n \text{Tr}\left(\mathcal{G}_{nn}\tau_1\right) \tag{145}$$

$$\mathcal{G}_{nm}^{-1} = \left[(i\omega - \Sigma + \mathcal{T}_0)\tau_0 + (\mathcal{T}_1 + F)\tau_1\right]\delta_{nm} \tag{146}$$

since $\Gamma_{nm} \propto \delta_{nm}\Gamma_n$ and $\Gamma_n \neq 0$ for $n = 1, ..., n_c$ and $\Gamma_n = 0$ for $n = n_c + 1, ..., N$. We have exactly the same kind of equations reported in Ref. [37]. Quite strikingly, in the so-called ergodic and long dwell-time regime, the authors of Ref. [37] solved Eqs. (144)-(146) for $N \gg n_c$ getting a Josephson current of the same form of Eq. (67), where the Thouless energy $E_T$ is replaced by $\Gamma^2/U \ll \Delta$. The crucial difference between the SYK$_2$ model discussed here, with respect to the case of a chaotic tunneling Josephson junction discussed in Ref. [37], is that the dot here is considered fully coupled with the leads, because all modes of the dot are equivalent to each other.

## 5.3 SYK$_q$ dot

Let us conclude discussing the generalized model with $q$ fermions. As shown previously, for $q = 2$ and fully contacted dot the effect of the hybridization and, more specifically, the induced pairing in the dot are very strong also in the large $N$ limit. For $q = 4$ instead the self-energy induced by the coupling with the leads remains the bare one for large $N$. One expect that the same situation is valid for any $q \geq 4$. In this situation we can directly write the large $U$ limit for the Josephson current as

$$I = \frac{\Gamma^2\Delta^2}{2\pi}\sin(\phi)\int_{-\infty}^{\infty} \frac{d\omega}{\Gamma^2\Delta^2\cos^2(\phi/2) - \left(G_0^{-1}(\omega)\sqrt{(\omega^2+\Delta^2)} + i\omega\Gamma_0\right)^2 - \omega^2\Gamma_3^2} \tag{147}$$

where we fixed $\mu = 0$ and, for $q \geq 2$, the exact Green's function in the large $N$ limit reads [40]

$$G_0^{-1}(\omega) = iC_q U^{\frac{2}{q}}\text{sgn}(\omega)|\omega|^{1-\frac{2}{q}} \tag{148}$$

with coefficient $C_q = \dfrac{(2\pi)^{1/q}\sec(\frac{\pi}{q})}{2\left[\left(1-\frac{2}{q}\right)\tan(\frac{\pi}{q})\right]^{1/q}\Gamma(1-\frac{2}{q})}$.

For any $q > 4$ and for weak tunneling, $\Gamma \ll \Delta$, we get the following result for the current

$$I \simeq \frac{\Gamma^2}{\pi}\sin(\phi)\int_0^\infty \frac{d\omega}{\Gamma^2\cos^2(\phi/2) + C_q^2 U^{\frac{4}{q}}\omega^{2-\frac{4}{q}}} \tag{149}$$

namely the superconducting pairing drops out completely, since $\Delta$ does not even play the role of an ultraviolet cut-off, as for $q = 4$. Introducing a cut-off $\Lambda$ the result, for any $q \geq 4$, is

$$I \simeq \frac{\sin(\phi)\,\Lambda}{\pi\cos^2(\phi/2)}\,{}_2F_1\left(\frac{q}{2(q-2)}, 1, 1 + \frac{q}{2(q-2)}; -\frac{C_q^2 U^{\frac{4}{q}}\Lambda^{2-\frac{4}{q}}}{\Gamma^2\cos^2(\phi/2)}\right) \tag{150}$$

where ${}_2F_1(a, b, c; z)$ is the hypergeometric function. For $q = 4$, we recover the result reported in Eq. (67) with $\Lambda = \Delta$, while for any $q > 4$ the limit $\Lambda \to \infty$ is finite therefore we do not need to use any cut-off.

For the very extreme case of $q \to \infty$ we have that $C_q \to 1/2$ and $G_0^{-1}(\omega) \to i\omega/2$, and always for $\Gamma \ll \Delta$, the current reduces simply to

$$I \simeq \Gamma\frac{\sin(\phi)}{|\cos(\phi/2)|} \tag{151}$$

namely, it approaches a $\pi$-junction upon increasing $q$, loosing the dependence also on $U$.

In the other limit, $\Gamma \gg \Delta$ and for $q$ such that $U^{2/q} \ll \Gamma$, e.g. $q \to \infty$, from Eq. (147) we get

$$I \simeq \frac{\Delta}{2}\frac{\sin(\phi)}{|\cos(\phi/2)|} \tag{152}$$

which is the same current for the non-interacting case, Eq. (119) for $t_o = 1$, at resonance.

# 6   Conclusions

We studied the Josephson effect obtained by contacting a $SYK_4$ dot by two superconducting leads. We showed that a proximity effect is induced in the dot, however the self-energy is weakly affected by the coupling with the leads in the so-called conformal limit, namely for large interaction and large number of particles. We found that, in this limit, the Josephson current is suppressed by $U$, the strength of the interaction, as $\ln(U)/U$ and becomes universal, since the current turns out to be independent on the superconducting pairing, and robust under phase fluctuations. This result implies that the Josephson current, at zero temperature, and in the conformal limit, is almost the same, up to logarithmic corrections, for all BCS-like superconductors. At finite temperature $T$, instead, the dependence on the superconducting gap is restored. The current becomes dependent on the ratio between the gap and the temperature and goes as $\Delta^2/T^2$ for sufficiently large temperatures. Finally we compare the Josephson current got for the $SYK_4$ with those obtained for other $SYK_q$ models.

### Acknowledgements

The author thanks R. Egger, D. Bagrets and N. Gnezdilov for useful comments and acknowledges financial support from the project BIRD 2021 "Correlations, dynamics and topology in long-range quantum systems" of the Department of Physics and Astronomy, University of Padova, from the European Union - Next Generation EU within the National Center for HPC, Big Data and Quantum Computing (Project No. CN00000013, CN1 Spoke 10 Quantum Computing) and from the Project "Frontiere Quantistiche" (Dipartimenti di Eccellenza) of the Italian Ministry for Universities and Research.

# A   Appendix

If we used Eq. (25) instead of Eq. (24) in Eq. (73), to take into account that the diagonal terms in the action Eq. (12) are null, we would get corrections of order $O(1/N)$ in the diagonal self-energy. Specifically, since $\delta\mathcal{G}_0 \propto \mathcal{G}_0(\mathcal{T}_1\tau_1 + \mathcal{T}_2\tau_2)\mathcal{G}_0$ where $\mathcal{G}_0$ contains only $\tau_0$ and $\tau_3$, then at the leading order in $1/N$ the self energy is corrected by $\delta\Sigma \propto \mathcal{G}_0^2 \delta\mathcal{G}_0$ which is only proportional to $\tau_1$ and $\tau_2$. Let us consider

$$\mathcal{G}_{nm}'^{-1} = i\omega\tau_0 + \mu\tau_3 - \Sigma - \frac{\delta\Sigma}{N} + \mathcal{T}_{nm}' \tag{153}$$

where $\delta\Sigma/N$ is the self-energy correction and $\mathcal{T}_{nm}'$ as in Eq. (25). We can define conveniently the following generic form for the Green's function

$$\mathcal{G}_0'^{-1} = \tilde{G}_0^{-1}\tau_0 + \tilde{G}_1^{-1}\tau_1 + \tilde{G}_2^{-1}\tau_2 + \tilde{G}_3^{-1}\tau_3 \tag{154}$$

where now

$$\tilde{G}_0^{-1} = i\omega - \Sigma_0 \tag{155}$$

$$\tilde{G}_1^{-1} = -\frac{1}{N}\left(\frac{\Gamma_1\Delta\cos(\phi/2)}{\sqrt{\omega^2 + \Delta^2}} + \delta\Sigma_1\right) \tag{156}$$

$$\tilde{G}_2^{-1} = -\frac{1}{N}\left(\frac{\Gamma_2\Delta\cos(\phi/2)}{\sqrt{\omega^2 + \Delta^2}} + \delta\Sigma_2\right) \tag{157}$$

$$\tilde{G}_3^{-1} = \mu - \Sigma_3 \tag{158}$$

so that Eq. (153) can be written as

$$\mathcal{G}_{nm}'^{-1} = \mathcal{G}_0'^{-1}\delta_{nm} + \mathcal{T}J_{nm} \tag{159}$$

where $\mathcal{T}$ is defined in Eq. (23), or, to simplify the notation,

$$\mathcal{T} = \frac{1}{N}\left(\mathcal{T}_0\,\tau_0 + \mathcal{T}_1\,\tau_1 + \mathcal{T}_2\,\tau_2 + \mathcal{T}_3\,\tau_3\right) \tag{160}$$

We can, now, employ Eq. (59) simply replacing $\mathcal{G}_0$ by $\mathcal{G}_0'$ and $\mathcal{G}$ by $\mathcal{G}'$, namely

$$\det[\mathcal{G}'^{-1}] = \left(\det[\mathcal{G}_0'^{-1}]\right)^{N-1}\left(\det[\mathcal{G}_0'^{-1}] + N\operatorname{Tr}[\mathcal{T}\mathcal{G}_0']\det[\mathcal{G}_0'^{-1}] + N^2\det[\mathcal{T}]\right) \tag{161}$$

getting, from Eqs. (154) and (160),

$$\det[\mathcal{G}'^{-1}] = \left(\det[\mathcal{G}_0'^{-1}]\right)^{N-1}\left[\left(\tilde{G}_0^{-1} + \mathcal{T}_0\right)^2 - \left(\tilde{G}_1^{-1} - \mathcal{T}_1\right)^2 - \left(\tilde{G}_2^{-1} - \mathcal{T}_2\right)^2 - \left(\tilde{G}_3^{-1} - \mathcal{T}_3\right)^2\right] \tag{162}$$

so that

$$\ln\det[\mathcal{G}'^{-1}] = (N-1)\ln\left[(\tilde{G}_0^{-1})^2 - (\tilde{G}_1^{-1})^2 - (\tilde{G}_2^{-1})^2 - (\tilde{G}_3^{-1})^2\right]$$
$$+ \ln\left[\left(\tilde{G}_0^{-1} + \mathcal{T}_0\right)^2 - \left(\tilde{G}_1^{-1} - \mathcal{T}_1\right)^2 - \left(\tilde{G}_2^{-1} - \mathcal{T}_2\right)^2 - \left(\tilde{G}_3^{-1} - \mathcal{T}_3\right)^2\right] \tag{163}$$

Expanding in $1/N$ we get

$$\ln\det[\mathcal{G}'^{-1}] = \ln[(\tilde{G}_0^{-1})^2 - (\tilde{G}_3^{-1})^2]^{N-1} + \ln\left[\left(\tilde{G}_0^{-1} + \mathcal{T}_0\right)^2 - \mathcal{T}_1^2 - \mathcal{T}_2^2 - \left(\tilde{G}_3^{-1} - \mathcal{T}_3\right)^2\right] + O\left(\frac{1}{N}\right)$$

$$= \ln\det[\mathcal{G}^{-1}] + O\left(\frac{1}{N}\right) \tag{164}$$

which is equal to the logarithm of Eq. (60) up to corrections of order $O(1/N)$.

Finally, we can invert a matrix of the form reported in Eq. (159), by using the geometric series,

$$
\mathcal{G}' = \left(I + \mathcal{G}_0' \mathcal{T} J\right)^{-1} \mathcal{G}_0' = \sum_{n=0}^{\infty} (-1)^n (\mathcal{G}_0' \mathcal{T} J)^n \mathcal{G}_0' \tag{165}
$$

and noticing that $J^n = N^{(n-1)} J$. We get the following result

$$
\mathcal{G}' = \left(I + \frac{1}{N} \sum_{n=1}^{\infty} (-1)^n (N \mathcal{G}_0' \mathcal{T})^n J\right) \mathcal{G}_0' = \mathcal{G}_0' I + \left(\tau_0 + N \mathcal{G}_0' \mathcal{T}\right)^{-1} \mathcal{G}_0' \mathcal{T} \mathcal{G}_0' J \tag{166}
$$

which can be rewritten explicitly as

$$
\mathcal{G}'_{nm} = \mathcal{G}_0' \, \delta_{nm} - \left[\left(N\mathcal{T} + \mathcal{G}_0'^{-1}\right)^{-1} \mathcal{T} \mathcal{G}_0'\right] J_{nm} \tag{167}
$$

useful to calculate the self-energy corrections due to the coupling with the leads.

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
