# Peer review of "Josephson current through the SYK model"

_SciPost Physics_

## Round 3 · Referee Report · Nikolay Gnezdilov (Referee 4) · 2024-7-8

Report

I have studied the resubmitted version of the paper. Even though the Author addressed multiple questions raised by other Referees and myself and extended the literature, unfortunately, I can not recommend the manuscript for publication in its current stage since the derivation of the paper's main results seems unclear. I explain my primary concern below.

In my previous report, I asked the Author about the large-$N$ structure of the saddle point of the model. At the beginning of Section 4, the Author writes the equation (50) that sets up the Green's function for the SYK quantum dot and is crucial for the paper's results. The second term in Eq. (50) originates from coupling the SYK quantum dot to the superconducting leads and is suppressed as $1/N$ if compared to the first term described by the bare SYK saddle-point equations (51,52).
The SYK saddle-point solution relies on the fact that only melon diagrams contribute to the SYK Green's function in the large-$N$ limit. For example, the diagram (I) in the attached figure contributes as ${\cal O} (1)$ if one accounts for the normalization of the interaction vertex $\propto 1/N^{3/2}$. The red dashed line is a disorder contraction and red letters $i, j, n, l$ denote the fermion flavors. However, the bare SYK Green's function may contain $1/N$ corrections to the $N$-independent saddle-point solution irrespective of coupling to the external leads. Such corrections to the bare SYK Green's function no longer appear as melon diagrams in every order. Naively, one can draw a diagram (II) that seems to contribute as ${\cal O}(1/N)$. These types of contributions are unaccounted for in Eq. (50), even though they are of the same order in $1/N$ as the one acquired due to coupling to the superconducting leads (second term in Eq. (50)).

In their reply to my previous report, the Author mentions that while the contribution to the SYK quantum dot Green's function due to coupling to the leads, indeed, scales as $1/N$, the determinant of the complete Green's function (${\rm Det} {\cal G}$) remains finite in the large-$N$ limit. Attributing this to Eq. (58) in the manuscript, I find this observation confusing since Eq. (58) extensively depends on $N$. To resolve this confusion, let me try to estimate the free energy since its phase derivative defines the Josephson current in Eq. (57) after all.
The free energy is
$$
{\cal F} = - \frac{1}{\beta} \sum_\omega \ln {\rm Det} {\cal G}^{-1}_{nm} = - \frac{1}{\beta} \sum_\omega \ln {\rm Det} ({\cal G}^{-1}_0 \delta_{nm} + {\cal T} J_{nm}),
$$
where ${\cal G}^{-1}_{nm}$, ${\cal G}^{-1}_0$ are defined in Eqs. (50-52) and ${\cal T}$ and $J_{nm}$ in Eqs. (22, 23). Here, I distinguish "${\rm tr}$" and "${\rm det}$" that correspond to the trace and determinant in the $2 \times 2$ Nambu space from "${\rm Tr}$" and "${\rm Det}$" that take care of the operations in $2 N \times 2 N$ space due to fermion flavors $n=\overline{1, N}$. Now, let me try to re-sum the logarithm in the flavor space:
$$
{\cal F} = - \frac{N}{\beta} \sum_\omega {\rm tr} \ln {\cal G}^{-1}_0 - \frac{1}{\beta} \sum_\omega {\rm Tr}\ln ( \delta_{nm} + {\cal G}_0{\cal T} J_{nm})
$$
$$
= - \frac{N}{\beta} \sum_\omega {\rm tr} \ln {\cal G}^{-1}_0 - \frac{1}{\beta} \sum_\omega \big( \sum_n {\rm tr} \, {\cal G}_0{\cal T} J_{nn} - \frac{1}{2} \sum_{n,m} {\rm tr} \, {\cal G}_0{\cal T} J_{nm} {\cal G}_0{\cal T} J_{mn} + \ldots \big)
$$
$$
= - \frac{N}{\beta} \sum_\omega {\rm tr} \ln {\cal G}^{-1}_0 - \frac{1}{\beta} \sum_\omega \big( N {\rm tr} \, {\cal G}_0{\cal T} - \frac{N^2 }{2} {\rm tr} \, {\cal G}_0{\cal T} {\cal G}_0{\cal T} + \ldots \big),
$$
where we used that the $J_{nm} =1$ for all $n,m = \overline{1,N}$. Since, ${\cal T} \propto 1/N$ as stated in Eq. (22), its $N$-dependence cancels $N, N^2, \ldots$ in the second term in the expression above. Thereby, we convert the expanded part of the free energy back to the logarithmic form:
$$
{\cal F} = - \frac{N-1}{\beta} \sum_\omega \ln {\rm det} {\cal G}^{-1}_0 - \frac{1}{\beta} \sum_\omega \ln {\rm det} ( {\cal G}^{-1}_0 + \widetilde{{\cal T}}),
$$
where we introduced $N$-independent $\widetilde{\cal T} \equiv N {\cal T}$. The second part of the free energy is $N$-independent. It contributes to the phase derivative and defines the Josephson current, seemingly in agreement with the Author's results.

The second term in the free energy is $1/N$ suppressed compared to the bare SYK free energy. One may expect this since the corresponding free energy term originates from accounting for the $1/N$ correction to the SYK saddle-point due to coupling to the superconducting leads. Yet, the omitted $1/N$ corrections to the bare SYK solution I mentioned above may contribute to the same order in the free energy here. Is it possible to clarify why this is not the case?

Attachment

Recommendation

Ask for major revision

  • validity: -
  • significance: -
  • originality: -
  • clarity: -
  • formatting: -
  • grammar: -

Author:  Luca Dell'Anna  on 2024-07-09  [id 4612]

(in reply to Report 1 by Nikolay Gnezdilov on 2024-07-08)
Category:
answer to question

I would like to thank Nikolay Gnezdilov for his careful reading of the new version of my paper.
Actually, more than presenting a criticism it seems to me that he provided a nice check and validation of the result reported in the paper.
Indeed, on the last issue about the 1/N expansion raised by the Referee I had been rather fast and unclear in my response.
As also shown by Gnezdilov in his report, the phase dependent part of the free energy does not depend on N (as shown in Eq. (59) of the paper), which generates the Josephson current calculated in the work.
Now, if we want to include 1/N corrections in the bare Green’s function, as suggested by the Referee, the phase independent part of the free energy acquires a term of order O(1) but does not contribute to the Josephson current (since it is phase independent) while the phase dependent part (which is O(1)), and therefore, the Josephson current, acquires trivially a term O(1/N) which is subleading and vanishes for large N. As a result, the Josephson current reported in the paper remains the same in the large N limit.

---

## Round 3 · Referee Report · Dmitry Bagrets (Referee 1) · 2024-7-18

Strengths

The manuscripts attempts to solve an interesting and timely problem.

Weaknesses

  1. Statistical properties of a coupling matrix between the SYK dot and superconducting leads are not analyzed properly.

  2. The time-reversal broken version of the SYK Hamiltonan leads to zero Josephson current after disorder averaging.

Report

I have carefully read the new versions of this paper and noticed some attempts to improve it. However, I cannot recommend it for publication in its present form.

In the revised version of the manuscript, my first concern related to the spin structure of the Hamiltonian was partially addressed. Nevertheless, the model of the SYK dot is still assumed to be spinless, which has crucial impacts on the final results (see below).

My second comment, regarding the random character of the coupling matrix elements between the dot and leads, was completely ignored. The author continues to use a fully symmetric and deterministic form of the coupling matrix J, which does not differentiate between different quantum numbers. From my perspective, this assumption is entirely wrong and contradicts the essence of SYK-like models. These models, by construction, describe disordered and chaotic strongly interacting systems. Therefore, a reasonable theoretical model must assume that the hybridization parameters t_pns are also random matrices.

The structure of Eq. (17) implies that the flux-dependent contribution of the T-matrix will average to zero. Specifically, the flux-dependent parts are weighted by the coupling rates (15, 16), which are complex numbers without definite signs. Assuming the most natural model, where the (complex) hopping amplitudes t_pns are Gaussian distributed, one concludes that the averages of (15) and (16) are strictly zero.

Therefore, the given spinless model of the SYK quantum dot cannot support a non-zero Josephson current (only mesoscopic fluctuations may remain). This is not surprising, as time-reversal symmetry is entirely broken in such a model.

To rescue this interesting project, I suggest the author discusses the spinful SYK Hamiltonian studied in:

[1] Hanteng Wang, A. L. Chudnovskiy, Alexander Gorsky, and Alex Kamenev, Phys. Rev. Research 2, 033025.

In this work, the SYK Hamiltonian with random two-body interactions has real spin-independent matrix elements, meaning that time-reversal symmetry is preserved. I hope that such a model of the SYK dot has the potential to provide a non-zero Josephson current after disorder averaging. However, I leave it to the author to decide whether to reconsider the calculations and resubmit the manuscript or to retract the current submission from SciPost Physics.

Requested changes

  1. Reconsider the model of the SYK dot along the lines outlined in the report and repeat calculations of the Josephson current.

Recommendation

Ask for major revision

---

## Round 3 · Author Response

Dear Editor, I would like to thank the Referees for their careful reading of my manuscript and for their useful comments.

Reply to the first Referee

I am indebted to Dimitry Bagrets for his accurate report and comments useful to improve the quality of my paper. I also thank him for saying that the manuscript attempts to solve an interesting and timely problem. In the new version of the paper I made several changes following his suggestions. Actually, instead of considering a modified spinful version of the SYK model, I preferred to include spin dependence on the coupling parameters, modifying the tunneling hamiltonian (Section 2.1) in the same spirit of what done for studying transport through a p-wave (spineless) superconductor contacted with s-wave superconductors (see Refs. [35, 36]). By doing that we assume both hopping and spin-flip processes occurring at the contacts. As correctly said by the Referee, the coupling is finite only if both spin projections are included in the coupling. The spin structure induces the presence of all the channels in Nambu space also for symmetric junctions. At the end of Sec. 2.1 I also discussed the possibility of including random fluctuations in the coupling. About the second comment, I devoted a new section, Sec. 4, to other cases: zero interaction, SYK_2 and SYK_q. In particular in Sec. 5.2 I derived the saddle point equations for the disordered non-interacting case which correspond to the rainbow diagrams as reported in Ref. [37] for a disordered Josephson junction. The difference with respect to that case is in the choice of the tunneling hamiltonian. At the end of Sec. 5.2 I discussed the case of Ref. [37]. What I showed is that the saddle point equations from replica path integral formulation are the same of those obtained from non-crossing diagrams solved in Ref. [37]. Finally, at the end of Sec. 4.1 I discussed the analogy between the result in Eq. (66) and that presented in Ref. [37] for ergodic and long-dwell time regime. Actually the two systems are different but the results are the same. I should confess that I was not aware of that result about disordered Josephson junction obtained by Brouwer and Beenakker. I really thank the Referee for pointing me out this relevant result.

Reply to the second Referee

I would like to really thank the Referee for carefully reading my manuscripts and for his/her useful comments. I thank the Referee also for finding the problem addressed by my paper well stated and interesting. I made several changes and additions in the new version of the paper in order to clarify the issues raised by the Referee. Moreover hereafter I reply point by point to the Referee’s questions.

1) In the effective action the terms proportional to N^3 come from the fact that one has to perform two sums for the corresponding auxiliary fields which are absent in the conventional decoupling (with a term proportional to N). Indeed, the self-energy Sigma is just a 2x2 matrix while the self-energies L and A are 2Nx2N matrices.

2) Although the factor 1/N in the tunneling matrix T, we can perform the limit N-> infinity in the current since the dimension of the T matrix also increases. As shown in the paper the Det(G) is finite for N-> infinity, providing that the coupling has that scaling factor. All the analytic results in the paper are meant for N-> infinity.

3) In the new version of the paper, at the end of Se. 2.1, I included a discussion about the random fluctuations of the coupling, and how to include them. Since the coupling is extensive one expects to have a sort of self-averaging effect. Moreover, as clarified in 2), the limit N->infinity can be performed (I do not assume finite N) therefore the replica diagonal solution, generally assumed for SYK model, is still valid.

4) In the new version in Sec. 5.2 I derived and showed that the saddle point equations from a replicated action correspond to the equations obtained from rainbow diagrams discussed in Refs. [37] and [38].

5) I perfectly agree with the Referee. Actually in the new version of the paper I included a spin structure in the coupling parameters, assuming both hopping and spin-flip processes in the junctions, as done for a p-wave (spineless) superconductor contacted with s-wave superconductors (see Refs. [35, 36]). Only including both spin projections the current can be finite.

6) I improved the bibliography.

Reply to the third Referee

I would like to thank the Referee for finding my work interesting. I made several changes and additions in the paper in order to better clarify the novelty of my work. The study of the effects of negative Hubbard terms or phonon bath would require a deeper analysis which goes well beyond the scope of my work. My intuition is that they could reinforce the induced pairing in the dot. In the introduction I included the following sentence: “Several attempts have been done also to include superconductivity in the SYK model [30–33] with the need of upgrading the original complex model to a spin-full version with, in addition, a mechanism of particle attraction provided by phonons or by a negative Hubbard term”. About the replica symmetric solution, it is justified since I assume N-> infinity limit, as in the uncoupled SYK model.

---

## Round 3 · List of Changes

Here the list of changes:

Abstract:
I added the following sentence: Finally we compared the results of the original four-fermion model with those obtained considering zero interaction, two-fermions and a generic q-fermion model.

Section 1:
In the Introduction I included the following sentences:
[…] Several attempts have been done also to include superconductivity in the SYK model [30–33] with the need of upgrading the original complex model to a spin-full version with, in addition, a mechanism of particle attraction provided by phonons or by a negative Hubbard term.
[…] The coupling between the dot and the superconductors involves uniformly all the degrees of free- dom of the dot and encodes either hooping and spin-flip processes, in the same spirit of what done linking a topological p-wave superconductor with s-wave superconducting leads [35, 36].
[…] Strikingly this result turns out to be formally the same as that obtained for a chaotic Josephson junction in the ergodic and long-dwell time regime reported in Ref. [37].
[…] Moreover we considered the SYK model with two fermionic operators, keeping the coupling with the leads the same. In this case the current de- pends on ∆ and is highly non sinusoidal. Finally for a generalized q-fermion SYK model we find that, for q > 4 and in the weak coupling regime, the current looses completely its dependence on the gap.

Section 2:
I modified Eq. (3) and added the following sentence afterwards:
This tunneling Hamiltonian encodes either the hopping which allows a fermion to jump into or out of the dot with same spin projection and the spin-flip processes at the interface for opposite spin projections, in the same spirit of Refs. [35], [36], where a topological superconductor made of spinless fermions is coupled to s-wave BCS superconducting electrodes.

Section 2.1:
I made several changes including a spin structure in the coupling parameters, Eqs. (8), (9), (11), (13-24). I included a new paragraph after Eq. (24) about random fluctuations of the coupling.

Section 4:
I modified Eq. (59) and added Eq. (60) and the following sentence: the coupling is finite for finite values of both t_up and t_down, namely there should be finite values of both spin projections, therefore also spin-flip processes in the presence of strongly polarized fermions in the dot, in order to have a non-vanishing Josephson current.
After Eq.(61) I added the following sentence: As a remark we point out that, using Eq. (24) instead of Eq. (23), we get the same expression for the current with subleading terms of order O (1/N ) (see Appendix A).

Section 4.1:
After Eq. (63) I added the following sentence: The term omega^2\Gamma_3^2 does not lead to any singularity since … which is exactly equal to Gamma^2, Eq.(60), and is positive defined.
After Eq. (66) I included a discussion comparing Eq. (66) with a similar result in Ref. [37]:
Quite interestingly a very similar result for the Josephson current reported in Eq. (66), had been obtained for a disordered Josephson junction in the so-called ergodic regime with long dwell time, namely when the ergotic time is small compare to \Delta^-1 and the Thouless energy scale E_T is such that E_T<<\Delta [37]. In our case Gamma^2/U plays the role of E_T, or alternatively Gamma^2/U the role of the dwell time, so that a large interaction U corresponds to a large diffusive time in the dot. We remind that, contrary to the case reported in Ref. [37], where the dot is a cavity weakly linked to the superconducting leads and the coupling with the leads involves few modes of the dot, in the present paper we considered a dot fully contacted to the leads, namely all the modes contribute to the coupling. The long dwell time in Ref. [37] is due to the diffusion of the particles in the cavity while, in our case, it is due to strong correlations which produces such a sort of self-trapping phenomenon. Moreover the anomalous self-energy plays a crucial role in Ref. [37], while in our case this term is negligible for large N because of the uniform coupling.

Section 5 is totally new.
The case with zero interaction has been shifted in Section 5.1. Eq. (90) is slightly modified because of the spin structure.

Section 5.2 on the SYK_2 model and Section 5.3 on the SYK_q model are totally new.

Section 6:
In the Conclusions I added the following sentence: Finally we compare the Josephson current got for the SYK4 with those obtained for other SYKq models.

Appendix A, about subleading terms and inversion of the Green’s function, is totally new.

---

## Editorial Decision

resubmitted